WHITE PAPER

# Neurocardiology: translational advancements and potential

N. Herring[1] [iD], O. A. Ajijola[2] [iD], R. D. Foreman[3], A. V. Gourine[4] [iD], A. L. Green[5] [iD], J. Osborn[6], D. J. Paterson[1] [iD], J. F. R. Paton[7] [iD], C. M. Ripplinger[8] [iD], C. Smith[9], T. L. Vrabec[10], H. J. Wang[11] [iD], I. H. Zucker[12] [iD] and J. L. Ardell[2] [iD]

[1]*Department of Physiology, Anatomy and Genetics, University of Oxford, Oxford, UK*
[2]*UCLA Neurocardiology Research Center of Excellence, David Geffen School of Medicine, Los Angeles, CA, USA*
[3]*Department of Biochemistry and Physiology, University of Oklahoma Health Sciences Center, Oklahoma City, OK, USA*
[4]*Centre for Cardiovascular and Metabolic Neuroscience, University College London, London, UK*
[5]*Nuffield Department of Surgical Sciences, University of Oxford, Oxford, UK*
[6]*Department of Surgery, University of Minnesota, Minneapolis, MN, USA*
[7]*Manaaki Manawa – The Centre for Heart Research, Department of Physiology, Faculty of Medical and Health Sciences, University of Auckland, Auckland, New Zealand*
[8]*Department of Pharmacology, University of California Davis, Davis, CA, USA*
[9]*Department of Physiology and Biophysics, Case Western Reserve University, Cleveland, OH, USA*
[10]*Department of Physical Medicine and Rehabilitation, School of Medicine, Case Western Reserve University, Cleveland, OH, USA*
[11]*Department of Anesthesiology, University of Nebraska Medical Center, Omaha, NE, USA*
[12]*Department of Cellular and Integrative Physiology, University of Nebraska Medical Center, Omaha, NE, USA*

Handling Editors: Harold Schultz & Kalyanam Shivkumar

The peer review history is available in the Supporting Information section of this article (https://doi.org/10.1113/JP284740#support-information-section).

**Abstract figure legend** Afferent signalling pathways of the autonomic nervous system and their efferent targets in the heart, kidneys and vasculature.

The Journal of Physiology

**Abstract**   In our original white paper published in the *The Journal of Physiology* in 2016, we set out our knowledge of the structural and functional organization of cardiac autonomic control, how it remodels during disease, and approaches to exploit such knowledge for autonomic regulation therapy. The aim of this update is to build on this original blueprint, highlighting the significant progress which has been made in the field since and major challenges and opportunities that exist with regard to translation. Imbalances in autonomic responses, while beneficial in the short term, ultimately contribute to the evolution of cardiac pathology. As our understanding emerges of where and how to target in terms of actuators (including the heart and intracardiac nervous system (ICNS), stellate ganglia, dorsal root ganglia (DRG), vagus nerve, brainstem, and even higher centres), there is also a need to develop sensor technology to respond to appropriate biomarkers (electrophysiological, mechanical, and molecular) such that closed-loop autonomic regulation therapies can evolve. The goal is to work with endogenous control systems, rather than in opposition to them, to improve outcomes.

(Received 6 March 2024; accepted after revision 3 September 2024; first published online 28 September 2024)

**Corresponding authors** N. Herring: Department of Physiology, Anatomy and Genetics, University of Oxford, Oxford, UK.    Email: neil.herring@dpag.ox.ac.uk. J. L. Ardell: UCLA Neurocardiology Research Centre of Excellence, David Geffen School of Medicine, Los Angeles, CA, USA.    Email: jardell@mednet.ucla.edu

## Introduction

This white paper starts by considering the major afferent inputs and cardiac efferent signalling pathways of the cardiovascular autonomic nervous system, and how they integrate in terms of reflex feedback control loops at different levels of the neuronal hierarchy (Fig. 1) to modulate cardiac function. The autonomic nervous system adapts and remodels in response to cardiovascular disease, which may be beneficial in the short term to maintain cardiac function and overall blood pressure, but in the long term can exacerbate the underlying pathophysiology and lead to disease progression. We review the implications of this specifically for arrhythmia induction, myocardial ischaemia/infarction, and chronic heart failure with an emphasis on the progress that has been made since the previous White Paper published in 2016 and incorporating more detail on the brainstem and higher centre control. Finally, we review the progress to date of different approaches to interventional autonomic regulatory therapy for cardiovascular disease, highlighting promising new directions and major challenges that need to be addressed. While therapeutic modulation of auto-

nomic function has not become part of mainstream practice beyond cardiac sympathetic denervation, better understanding and new technologies are contributing to translational potential. These include electrode design and non-invasive and chemical denervation to specific organs. Important updates since the 2016 White Paper relate to the use of novel techniques to stimulate the autonomic nervous system, and use of denervation techniques to modulate cardiovascular and renal function in disease states.

## Structural and functional organization of the cardiac nervous system: afferent signalling

**Cardiac afferent neurons.** Cardiac afferent neurons can be categorized into three main types: (1) those that sense mechanical stimuli, (2) those that detect chemical signals, and (3) those that can respond to both mechanical and chemical cues (i.e. bimodal) (Armour, 2004; Foreman, 1999; Fu & Longhurst, 2009; Kember et al., 2001; Malliani & Lombardi, 1982; Thompson et al., 2000; Thorén, 1977; Thorén et al., 1976). These sensory neurons are found in various locations, including the nodose and dorsal root

**Neil Herring** (left) completed his PhD and medical degree at the University of Oxford before training in cardiology. He was awarded Intermediate and Senior Research Fellowships from the British Heart Foundation and now leads a translational research group alongside clinical work at the Oxford Heart Centre as a Professor of Cardiovascular Medicine. He is a tutorial fellow in medical science at Exeter College, Oxford. **Jeffrey L. Ardell** (right) completed his BA at Colorado College and PhD at the University of Washington, Seattle, before being appointed as a Professor in the College of Medicine at East Tennessee State University. In 2014 he became the Founding Director and Professor of Medicine and Anesthesiology at the UCLA Neurocardiology Research Program of Excellence and UCLA Cardiac Arrhythmia Centre. Both Neil and Jeff are part of the Leducq International Network on *Bioelectronmics for Neurocardiology*.

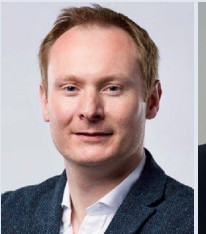
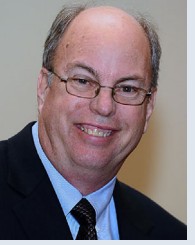

ganglia (DRG) (Armour et al., 1994; Hoover et al., 2008; Vance & Bowker, 1983) as well as extracardiac (Armour, 1983, 1986a) and intrinsic cardiac ganglia (Ardell et al., 1991; Beaumont et al., 2013a). The arrangement of afferent, efferent and interneurons and their architecture in intrinsic ganglionic plexi in the epicardial fat pads was first referred to as the 'heart's little brain' in the pioneering work of J Andrew Armour (Armour, 2008). Modern tissue clearing techniques combined with high throughput screening PCR and immunohistochemistry have started to uncover a high level of multi-chamber neuronal complexity within ganglionic plexi of the mouse (Rajendran et al., 2019), pig and human hearts (Hanna

et al., 2021) and their functional interactions are only just starting to be delineated.

Within the heart is an extensive network of myelinated and non-myelinated spinal sensory nerve fibres that connect to somata in the DRG located in the upper thoracic spinal cord (T1-T6). They are sensitive to various substances such as hydrogen ions (Uchida & Murao, 1975), potassium (Meller & Gebhart, 1992; Webb et al., 1983), oxygen radicals (Huang et al., 1995; Ustinova & Schultz, 1994a), bradykinin (Uchida & Murao, 1974), adenosine (Arora & Armour, 2003), ATP (Katchanov et al., 1996), and arachidonic acid metabolites (Fu et al., 2008; Nerdrum et al., 1986; Staszewska-Barczak, 1983; Sun

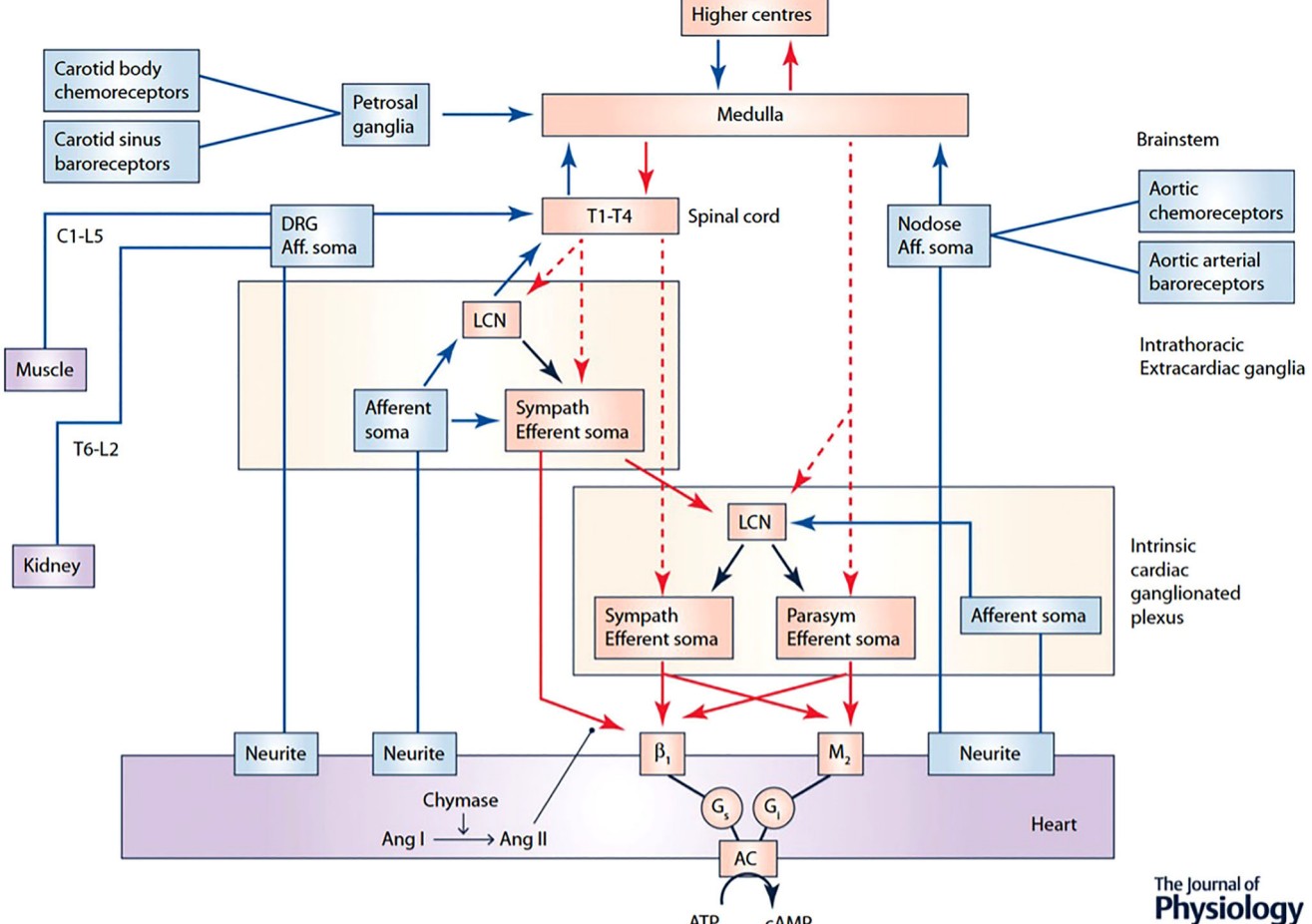

**Figure 1. Network interactions occurring within and between peripheral ganglia and the central nervous system for autonomic control of the heart**
The cardiac nervous system is composed of multiple (distributed) processing centres from which independent and interdependent neural feedback and feed-forward neural circuits interact to control regional cardiac electrical and mechanical function. Afferent projections are indicated in blue and efferent projections in red (dashed lines, preganglionic; continuous lines, postganglionic). The intrinsic cardiac nervous system (ICNS) possesses sympathetic (Sympath) and parasympathetic (Parasym) efferent postganglionic neurons, local circuit neurons (LCN) and afferent neurons. Extracardiac intrathoracic ganglia contain afferent neurons, LCN and sympathetic efferent postganglionic neurons. Neurons in intrinsic cardiac and extracardiac networks form nested feedback loops that act in concert with CNS feedback loops (spinal cord, brainstem, hypothalamus and forebrain) to coordinate cardiac function on a beat-to-beat basis. These systems demonstrate plasticity which underlies adaptations to acute and chronic stressors. Ang I, angiotensin I; Ang II, angiotensin II; AC, adenylate cyclase.

et al., 2001). While they are primarily chemosensitive, they can also respond to mechanical changes, particularly during intense ventricular contractions (Malliani et al., 1983). It is possible to disrupt nerve fibre transmission responsible for reflex responses to bradykinin and nicotine by applying phenol solutions to the heart's surface, particularly around the atrioventricular groove (Inoue et al., 1988; Ito & Zipes, 1994), although a significant portion of these fibres run close to the surface of the left ventricle (Brändle et al., 1994; Inoue et al., 1988; Ito & Zipes, 1994; Wang et al., 2014; Zucker et al., 1995b). Emerging techniques like tissue clearing may facilitate the creation of a comprehensive cardiac map detailing their distribution.

Other cardiac sensory neurons are found in nodose and thoracic DRGs (Armour, 2008). These sensory endings transmit various chemicals, including neuropeptides like substance P, bradykinin, and calcitonin gene-related peptide, potentially initiating local inflammatory, vascular and permeability changes through axon reflexes (Franco-Cereceda et al., 1993; White et al., 1993; Yaoita et al., 1994). Whilst chronic activation of these sensory endings in cardiovascular diseases may not always produce symptoms (Foreman, 1999), they could play a vital role in initiating cardiac remodelling (Janig, 2014; Wang et al., 2014; Yoshie et al., 2020).

The transmission of information from primary DRG sensory fibres in the heart to thoracic spinal neurons' dorsal horn primarily involves glutamatergic signalling (Foreman, 1999; Oliveira et al., 2003), often supplemented by substance P (Ding et al., 2008a; Hua et al., 2004). Increased substance P release may be observed during coronary artery occlusion, especially in the dorsal laminae I and II and deeper laminae (Hua et al., 2004). This release is significantly impacted by T2-T5 dorsal root trans-ections and involves transient receptor potential vanilloid 1 (TRPV1) channels – a mechano-, temperature, and pH-sensitive, calcium permeable channel that is activated by capsaicin (Guo et al., 2007; Steagall et al., 2012).

The pathways through which cardiac sympathetic afferent neurons participate in pain and sympathetic responses are complex. They may partly ascend through the dorsal columns, spinothalamic and spinoreticular tracts to cortical terminations (Foreman, 1999) and may project to mid and hindbrain autonomic integration areas. Cardiac afferent neurons influence neuronal activity in the paraventricular nucleus (PVN) (Affleck et al., 2012; Reddy et al., 2005; Xu et al., 2013) and nucleus tractus solitarius (NTS) (Wang et al., 2006, 2007) where they play a role in modulating the arterial baroreflex and chemoreflex (Chen et al., 2015; Gao et al., 2007; Reddy et al., 2005; Wang et al., 2007). Activation during coronary ischaemia can initiate lethal arrhythmias (Fukuda et al., 2015), which can be mitigated by thoracic dorsal rhizotomy (Schwartz et al., 1976) or stellectomy (Bourke et al., 2010; Vaseghi

et al., 2014). While older data show marked remodelling of vagal afferents characterized by a loss of arborization, especially in the atria, in the setting of chronic heart failure (Zucker et al., 1977, 1979), activation of cardiac afferents is also central to initiating more global neural remodelling (Wang et al., 2014; Zucker et al., 2012), characterized by changes in neuronal morphology, connectivity and electrical properties.

Mechano and chemosensitive afferents reside in various layers of the atria and ventricles. Their roles in the reflex control of autonomic function (e.g. sympatho-inhibition or sympatho-excitation) depend on central processing through either the vagus or thoracic dorsal root ganglia. Most of the evidence suggest that receptors located in chambers that are highly distensible and low in pressure mediate mechanosensitive sympatho-inhibition (Zucker, 1991). There are human data showing deficits in the reflex responses of decreased blood volume to the heart (without changes in arterial pressure) that are presumed to be the result of deficiencies in the function of cardio-pulmonary receptors in several disease states including heart failure (Mohanty et al., 1989) and atrial fibrillation (Malik et al., 2019, 2022). On the other hand, ventricular receptors appear to be more chemosensitive in response to ischaemic byproducts and signal through the spinal cord to be sympatho-excitatory, although it should be pointed out that many of these sensory endings are multimodal, both mechano and chemosensitive (Yu, 2023).

Nodose afferents transmit signals via the vagus nerve to the NTS (Bourke et al., 2010; Brändle et al., 1994; Chen et al., 2015; Chernicky et al., 1984). These neurons are sensitive to mechanical deformation in the atria and ventricles but can also transduce chemical changes during coronary ischaemia that persist for up to an hour, with purinergic receptors playing a crucial role in this sensitization (Jin et al., 2004; Wan et al., 2010).

In terms of overall control, reflex sympathetic output to the heart and vascular system are derived from two afferent inputs: (1) those that reach the brainstem at the NTS and (2) those that project to spinal cord neurons, distinguishing between cranial visceral afferent neurons (like those in the nodose ganglia) and spinal dorsal root ganglia. In general, cranial neural-activated pathways tend to dampen cardiac function through negative feed-back mechanisms, while spinal visceral afferent neurons trigger harmful positive feedback to sympathetic effector neurons. Activation of cardiac afferent input to the NTS is usually not consciously perceived, whereas stimulation of cardiac spinal afferent inputs can lead to perceptible pain and indirect pathways that influence brainstem autonomic centres.

**Pulmonary afferent neurons.** Pulmonary afferent neurons receive signals from both vagal and spinal pathways. Previous research has focused on the impact of

pulmonary vagal neurons on respiratory and upper airway function and regulation. Stimulation of vagal sensory fibres has been found to influence breathing rate and tidal volume through the Hering-Breuer reflex (Schelegle, 2003; Schelegle & Green, 2001; Vadhan & Tadi, 2024; Wyman, 1976). The bradycardia observed in response to lung inflation and to inhalation of noxious stimuli is thought to be mediated by afferents of vagal origin. Although vagal-derived afferents are the predominant sensory fibres within the lung, a number of studies have identified pulmonary spinal afferents using retrograde labelling (Kummer et al., 1992; Springall et al., 1987). Tracheal afferent nerve fibres originate in the C1 and T1-T4 DRG (Dalsgaard & Lundberg, 1984; Kummer et al., 1992) and pulmonary spinal afferent activity can be recorded from T2-T4 (Kostreva et al., 1981), the same upper thoracic segments through which cardiac sympathetic afferents pass (Wang et al., 2014). Whilst spinal chemically sensitive sympatho-inhibitory afferent fibres have been identified in the rabbit (Soukhova-O'Hare et al., 2006), two recent studies (Adam et al., 2019; Shanks et al., 2018) indicate that activation of pulmonary spinal afferents modulates an excitatory pressor response in anesthetized, vagotomised rats, resulting in increased heart rate, blood pressure and renal sympathetic nerve activity. This reflex can be blocked with epidural delivery of the selective afferent neurotoxin resiniferatoxin (RTX) at the level of T1-T4 DRGs, confirming an overlap convergence of thoracic DRGs between cardiac and pulmonary spinal afferents.

**Arterial baroreceptor.** Eighty-six years ago, Heymans described the carotid sinus nerve that mediates the baroreceptor reflex and the response from the carotid body chemoreceptors, for which he won the Nobel Prize in 1938. Credit too must also go to Hering who in the 1920s reported baroreceptors in the carotid sinuses that, when stimulated, evoked bradycardia and hypotension (de Castro, 2009). These receptors are mechano-sensitive nerve endings located bilaterally in the carotid sinuses and aortic arch embedded in specialized regions, which contain a greater proportion of elastin. The cell bodies of baroreceptor afferent neurons are in the nodose (aortic) and petrosal (carotid sinus) ganglia that have central projections to the intermediate region of the NTS (Spyer, 1994). Baroreceptors consist of myelinated A and unmyelinated C-fibres with different pressure thresholds and firing dynamics. Fast conduction fibres (pressure threshold around 60 mmHg) exhibit dynamic properties responding to pulse pressure and its rate of rise with higher sensitivity and narrower operating pressure ranges relative to slowly conducting fibres; the latter have a higher-pressure threshold ($\sim$130 mmHg in rat; (Thoren et al., 1983) and can fire spontaneously, encoding changes in mean arterial pressure (Seagard et al., 1990).

Subsequently, A and C fibres were found to control baroreflex sensitivity and resting levels of arterial pressure, respectively (Seagard et al., 1990; Seagard et al., 1993). Interestingly, as the arterial wall stiffens, baroreceptor pressure threshold increases becoming less sensitive to pressure (Thoren et al., 1983) which may cause the arterial pressure set point to rise. Interestingly, arterial baroreceptors are also sensitive to changes in blood flow detected within the carotid sinuses by a subset of afferents termed rheoreceptors (Hajduczok et al., 1988). If blood flow is elevated at constant pressure, this augments carotid sinus baroreceptor activity and decreases the pressure threshold of activation through increased shear stress.

The molecular basis of pressure sensing by baroreceptor afferents has been scrutinized and several mechanisms are purported to contribute. One of these is the acid sensing ion channel ASIC2, which is a subfamily of the DEG/ENaC superfamily (Lu et al., 2009). In ENaC knockout mice the baroreflex was depressed and animals developed hypertension. More recently, Piezo1 and Piezo2 mechanically activated channels were found in baroreceptor afferent neurons (Zeng et al., 2018), where optical stimulation of afferent neurons expressing these channels produced a baroreflex type response.

The baroreceptor reflex can be tested using the Oxford method of administering intra-venously vasoactive compounds (sodium nitroprusside and phenylephrine) to lower and raise blood pressure, respectively, while measuring reflex changes in heart rate and/or sympathetic motor output. This produces a sigmoidal relationship forming a baroreflex function curve where the linear portion of the curve reveals baroreflex sensitivity that falls across a pressure range. Pressure threshold and pressure saturation of the reflex function curve can be measured, as can the operating point defined as the baroreflex sensitivity at basal arterial pressure. Alternatively, the 'spontaneous' baroreceptor reflex gain can be computed from spontaneous changes in arterial pressure (up and down ramps in pressure) that occur at a specified number of beats before the change in heart rate or sympathetic activity. This has the advantage of avoiding vasoactive compounds, which in themselves may alter baroreflex sensitivity (Casadei & Paterson, 2000).

Stimulation of arterial baroreceptors by raising arterial pressure evokes a powerful reflex response including inhibition of sympathetic activity, bradycardia, a reduction in arterial pressure, suppression of both ventilation and vasopressin, and bronchodilatation. One unique characteristic of activation of baroreceptors is that they produce an antagonistic action on the sympathetic *versus* the parasympathetic nervous systems; whilst most other visceral reflexes co-activate or co-inhibit these motor outputs (Paton et al., 2005). Baroreflex sympathoinhibition tends to occur at a lower pressure threshold than the vagal cardio-inhibitory response

(Simms et al., 2007). One possibility is that these different responses originate from specific baroreceptor sites: carotid sinus *versus* aortic baroreceptors, at least in the rat (Pickering et al., 2008).

Physiologically, during exercise, there is a rapid resetting of the baroreflex to allow arterial pressure to rise. This is mediated by an inhibitory GABAergic mechanism within the NTS (Potts et al., 2003) driven by activation of afferents embedded in the skeletal muscle when it contracts. Additionally, during the defence reaction (fight or flight), the baroreceptor reflex is reset via GABA$_A$ receptors in the NTS and driven by descending pathways from the hypothalamus (Jordan et al., 1988). During both exercise and during 'fight or flight,' the baroreflex remains sensitive operationally but is reset over a higher-pressure range (Dampney, 2017). In contrast, in conditions of hypertension, the baroreflex function curve has diminished sensitivity in animals (Simms et al., 2007) and humans (Johansson et al., 2005). Whether harnessing ways to modulate baroreceptors can be used effectively to control blood pressure will depend on whether they provide either short- or long-term regulation of arterial pressure; this has been controversial and is considered in a separate section below.

**Arterial chemoreceptors.** Arterial chemoreceptors are found on the aortic arch (aortic bodies) and bilaterally at the bifurcation of the common carotid artery (carotid bodies; CB). They consist of groups of specialized glomus cells (type I) that are closely associated with smaller sustentacular or type II cells. Aortic bodies are innervated by vagal afferents with cell bodies in the nodose ganglion, whereas petrosal afferents provide sensory innervation to the CBs. Both sensory afferents project to the NTS. Glomus cells release a host of transmitter substances including acetyl choline, adenosine triphosphate (ATP), noradrenaline and dopamine that act on receptors located on either primary afferent terminals and/or adjacent glomus and sustentacular cells and may therefore integrate afferent information prior to generating an output response.

The CBs are one of the most vascular organs in the body, which ensures sensitive peripheral chemoreception at the major arteries to the brain. They have both sympathetic and parasympathetic innervation, which influence local control of blood flow and potentially have a direct effect on glomus cells, given that both express $\alpha_1$-adrenoceptors (Felippe et al., 2023). Reducing blood flow via sympathetically mediated vasoconstriction at the onset of exercise or from reduced cardiac output in heart failure may increase their sensitivity (Ding et al., 2011).

Classically, peripheral chemoreceptors respond to hypoxia, hypercapnia and low pH. There are several theories behind the mechanisms of oxygen sensing by the CBs that are not necessarily mutually exclusive.

One suggestion is that hypoxia inhibits mitochondrial respiration and electron transport (Sommer et al., 2016; Sommer et al., 2020), resulting in electron leakage from complex III and/or complex I, where free radical generation and an excess of NADH act as signalling molecules by inhibiting membrane Twik-related acid-sensitive K channels 1 and 3 (TASK1/TASK3) (Buckler, 2015). This then causes depolarization and activation of voltage-gated calcium channels to produce neurosecretion (Lopez-Barneo et al., 2016; Moreno-Dominguez et al., 2020; Sommer et al., 2020). Roles for haemoxygenase, carbon monoxide and hydrogen sulphide have also been reported, especially during intermittent hypoxia (Prabhakar et al., 2018). Interestingly other bloodborne stimulants have been described, including hyperkalaemia (Prabhakar et al., 2018), hypoglycaemia (Garcia-Fernandez et al., 2007), leptin (Shin et al., 2019), insulin (Baby et al., 2023) and most recently glucagon-like-peptide 1(Pauza et al., 2022). This suggests that the CB is a multi-modal receptor that senses a plethora of signals and plays a role in homeostatic regulation of the cardiovascular, respiratory and metabolic function (Thakkar et al., 2023).

The peripheral chemoreceptors mediate reflex responses that affect respiration, arterial pressure, heart rate, levels of hormones (adrenaline, adreno-corticotropic hormone and vasopressin) and bronchomotor tone. A primary chemoreflex response to hypoxia includes hyperventilation, bradycardia, a rise in arterial pressure, sympathoexcitation and bronchoconstriction. However, as ventilation ensues, Hering-Breuer afferents (lung inflation) are recruited that themselves produce a secondary reflex including inspiratory off-switching, tachycardia and sympathoinbition, causing blood pressure to fall. These changes oppose the primary response and the net results depends on the intensity of the stimulus, the degree of lung inflation and species (Marshall, 1994).

Most recently the structure/function and reflex components of the CB have generated a novel hypothesis on CB-to-brain communication where individual afferent signalling lines from distinct groups of glomus cells project to different central reflex arcs. This is known as the 'ribbon cable' hypothesis and may explain how different stimuli acting on the CB could produce unique and appropriate patterns of reflex response (Zera et al., 2019). This concept, if true, will be crucial for permitting therapeutic pharmacological targeting of the CB such that sympathetic vasomotor activity could be dampened without affecting chemoreflex control of ventilation.

**Renal afferent neurons.** Sensory nerves are associated with all branches of the renal arteries in varying density whereas the renal veins are more sparsely innervated (Marfurt & Echtenkamp, 1991). The sensory innervation of the renal cortex, particularly afferent and efferent

arterioles is prominent (Ditting et al., 2009). Most glomeruli are closely associated with sensory fibres in the mouse (Tyshynsky et al., 2023); however, their functional role has not been studied. The sensory innervation of tubular structures has not been well characterized. CGRP-positive nerve fibres are associated with tubules and appear to terminate in the interstitial space between tubules (Ditting et al., 2009). The pelvis has the highest density of sensory innervation relative to other structural components of the kidney and has been shown to express substance P, nitric oxide synthase and TRPV1 channels (Kopp et al., 2001).

Renal sensory nerves have cell bodies located in lower thoracic and upper lumbar DRG. The contribution of nodose ganglion neurons to renal sensory innervation has also been reported (Gattone et al., 1986), suggesting renal afferents may project to the brain via the vagus nerve (Cheng et al., 2022). Anatomical and physiological studies have demonstrated that renal primary afferent neurons terminate in the spinal cord and brainstem (Ammons, 1986; Ciriello & Calaresu, 1983; Knuepfer et al., 1988; Kuo et al., 1983; Simon & Schramm, 1983; Wyss & Donovan, 1984). Interestingly, immunolabelling for cFos as a marker for neuronal activation in the spinal cord following occlusion of the renal artery or vein suggests that these two perturbations engage distinct dorsal horn circuits for processing of renal sensory input (Rosas-Arellano et al., 1999). cFos labelling has also been employed to identify neurons in the NTS, rostral ventrolateral medulla (RVLM), supraoptic nucleus (SON) and PVN activated during electrical stimulation of renal nerves or perfusion of the renal pelvis with hypertonic saline (Goodwill et al., 2017; Solano-Flores et al., 1997).

Renal sensory nerves have typically been categorized as mechano- or chemosensitive (Kopp, 2015; Stella & Zanchetti, 1991). Single-unit recordings have revealed mechanoreceptors differentially sensitive to changes in renal arterial and venous pressure, as well as physiologically relevant changes in pelvic pressure (Genovesi et al., 1993; Kopp, 2015). Two types of renal chemoreceptors have been described in the rat. Type R1 chemoreceptors respond to extreme ischaemia, whereas R2 are spontaneously active and increase their firing rate in response to changes in ionic composition. Activation of afferent renal nerve activity in response to epithelial sodium channels (ENaC) modulates central autonomic pathways (Goodwill et al., 2017). Renal sensory neurons are also able to respond to low pH, reinforcing the significance of chemosensation in the kidney (Ditting et al., 2009; Recordati et al., 1980).

Stimulation of sensory renal nerves by reduced renal perfusion and electrical stimulation results in a variety of sympathetically mediated responses including increased arterial pressure, heart rate and total peripheral vascular resistance (Ashton et al., 1994; Faber & Brody, 1985;

Stella et al., 1987). Although activation of sensory renal nerves typically increases sympathetic activity to various cardiovascular targets, it has been reported to suppress sympathetic activity to the kidneys in the anaesthetized rat (Colindres et al., 1980; Zanchetti et al., 1984).

**Muscle afferent neurons.** Metabolic stimulation and mechanical deformation of contracting muscle stimulate sensory nerve endings and induce global sympatho-excitation, resulting in the 'exercise pressor reflex' (Alam & Smirk, 1937; Mitchell & Smith, 2008; Mitchell et al., 1983; Smith et al., 2006). Mechanical events in the musculature are transduced by mechanoreceptors associated with thinly myelinated group III fibres (Kaufman et al., 1983; Kaufman et al., 1984). Local metabolic byproducts are transduced by afferent neurons with unmyelinated (group IV) axons (Kaufman et al., 1983; Kaufman et al., 1984). Because these muscle receptors display polymodal transduction, exercise-induced sensory activation can elicit substantial changes in autonomic neuronal cardio-respiratory adjustments. Signals from muscle afferents in humans can influence the activity of key autonomic ponto-medullary regions of the brainstem such as the NTS, RVLM, caudal ventrolateral medulla, lateral tegmental field, nucleus ambiguous (NA) and the ventromedial region of the rostral periaqueductal grey (PAG) (Iwamoto & Kaufman, 1987; Iwamoto et al., 1982; Li et al., 1997). Such sensory reflex activation preferentially increases sympathetic drive to the heart compared to the peripheral vasculature such that pressor responses are primarily due to increasing cardiac efferent neuronal output concomitant with central blood volume mobilization. This may act as an important feedback mechanism in the cardiovascular response to exercise and help calibrate future feedforward responses at the start of exercise.

## Structural/functional organization of cardiac nervous system: cardiac motor neurons

**Cardiac sympathetic efferent neurons.** Presympathetic efferent neurons originate in the brainstem and communicate with cardiac sympathetic preganglionic neurons in the intermediolateral cell column of the cervical and cranial thoracic region of the spinal cord (Guyenet et al., 2013). Preganglionic neurons then project to sympathetic efferent postganglionic neurons in the stellate ganglia and the first few thoracic ganglia of the sympathetic chain (Buckley et al., 2016). Innervation of atria and ventricles originates from both left and right sided ganglia, although a degree of laterality has been observed in some species (Ajijola et al., 2013; Ardell et al., 1988; Dacey et al., 2022; Vaseghi et al., 2012). There are also sympathetic efferent postganglionic neurons

within the intrinsic cardiac ganglionated plexi that can influence electrical and mechanical indices in both atria and ventricles (Butler, Smith, Cardinal et al., 1990; Butler, Smith, Nicholson et al., 1990; Cardinal et al., 2009). However, if one or more intrinsic cardiac ganglionated plexuses are compromised, overall functional control can remain largely preserved (Leiria et al., 2011; McGuirt et al., 1997; Randall et al., 1998, 2003).

**Cardiac parasympathetic efferent neurons.** Cardiac parasympathetic efferent preganglionic neurons arise within the medulla oblongata (e.g. primarily NA) (Dergacheva et al., 2014; Hopkins & Armour, 1982; Massari et al., 1995) and project to efferent postganglionic neurons distributed throughout the cardiac ganglionated plexuses via the left and right vagi. Recent ultrasound guided microneurography techniques have allowed direct recording of vagal activity in humans (Ottaviani et al., 2020), which shows both cardiac and respiratory modulation (Patros et al., 2022). This is a mixed readout of both efferent activity as well as the 70%–80% afferent fibres carried by the vagi. Efferent postganglionic neurons in intrinsic cardiac ganglia influence the electrical and mechanical function of both atria and ventricles (Arora et al., 2003; Cardinal et al., 2009; Yuan et al., 1994). The ganglionic plexi receive bilateral vagal preganglionic inputs (Ardell & Randall, 1986; McGuirt et al., 1997; Yamakawa et al., 2014), and within the ganglionic plexuses further processing occurs via local circuit neurons (LCNs) (Beaumont et al., 2013b; Rajendran et al., 2016; Salavatian et al., 2016). The intrinsic cardiac nervous system can be viewed as a 'final common pathway' for integrated reflex control of regional cardiac electrical and mechanical function (Armour, 2008; Kember et al., 2011) and is able to function even when chronically decentralized from higher centres (e.g. following heart transplant) (Ardell et al., 1991; Murphy et al., 2000; Vaseghi et al., 2009).

**Neuron-myocyte interface.** Whilst the heart is richly innervated by both sympathetic and parasympathetic efferent neurons, nerve fibre density is non-uniform. The atria typically have a higher density of sympathetic nerve fibres compared to the ventricles, with abundant innervation of the sinoatrial and atrioventricular nodes (Crick et al., 1996; Crick, Sheppard et al., 1999; Kawano et al., 2003). There is also denser sympathetic innervation at the base of the heart compared to the apex and at the epicardium compared to endo- and subendocardial regions (Crick, Anderson et al., 1999; Ito & Zipes, 1994; Kawano et al., 2003; Wang et al., 2020). Despite this non-uniformity, detailed microscopy studies have demonstrated that nearly every cardiomyocyte is in contact with one or more sympathetic processes (Freeman et al., 2014; Zaglia et al., 2013). Parasympathetic nerve

fibres were historically assumed to have a higher density in the atria compared to the ventricles (Coote, 2013). However, significant ventricular parasympathetic innervation has been demonstrated (Ito & Zipes, 1994), with ventricular density of parasympathetic fibres exceeding atrial density in porcine hearts (Ulphani et al., 2010), whereas human hearts have greater abundance of parasympathetic fibres in the atria *vs*. ventricles (Kawano et al., 2003). Unlike sympathetic gradients, parasympathetic innervation appears to be denser in the endo- *vs*. epicardium (Ulphani et al., 2010).

Sympathetic processes demonstrate a prototypical 'pearl necklace' morphology, with the 'pearls' corresponding to varicosities from which neurotransmitters are released (Prando et al., 2018). Co-culture studies have revealed a close proximity of varicosities and cardiomyocytes of the order of <100 nm (Landis, 1976). Within the sympathetic neurocardiac junction, noradrenaline (norepinephrine) release is diffusion restricted, leading to local noradrenaline concentrations within the cleft of the order of ∼100 nM (Prando et al., 2018). Although noradrenaline release is responsible for prototypical sympathetic responses within the heart (positive inotropy, lusitropy and dromotropy) primarily via activation of $\beta$1-adrenergic receptors, detailed assessment of cardiomyocyte responses in co-culture has revealed that only those $\beta$ receptors directly adjacent to neuronal varicosities become activated (Prando et al., 2018). This local $\beta$ receptor signalling leads to locally restricted activation of cyclic-AMP (cAMP) within the cardiomyocyte, (Prando et al., 2018) potentially contributing to the sub-cellular compartmentalized cAMP signalling that is known to occur and is critical for cellular signalling (Bers et al., 2019).

The presence of sympathetic neurons also appears to promote the development of specialized signalling domains within the adjacent cardiomyocyte. These domains are enriched with $\beta$1-Ars and the scaffold proteins SAP97 and AKAP150 but are devoid of $\beta$2-Ars (Shcherbakova et al., 2007). Such signalling scaffolds were not present when cardiomyocytes were cultured alone, suggesting neuro-cardiac interaction is necessary to form these adrenergic signalling complexes. Other studies have revealed that adrenergic stimulation of cardiomyocytes leads to increased expression of gap junction proteins (connexin-43) and increased cell–cell coupling (Salameh et al., 2006). Cardiac sympathetic neurons have also been shown to be a strong regulator of cardiomyocyte size via $\beta$2-AR-mediated effects on proteolysis, with increased cell size associated with increased nerve density transmurally across the wall of rodent and human hearts (Pianca et al., 2019; Zaglia et al., 2013). Interestingly, there is some speculation that signalling at the neuron–myocyte interface can also work in reverse (i.e. the adjacent neuron can also respond to cardiomyocyte cues). Nerve growth

factor (NGF) plays a significant role in neuronal growth and survival and cardiomyocytes can secrete mature NGF (although at very low levels) (Bierl et al., 2005; Kaye et al., 2000; Zhou et al., 2004). Moreover, vesicular NGF release may depend on cAMP activation (Edwards et al., 1988), indicating that reciprocal crosstalk between myocytes and sympathetic neurons is possible (i.e. activation of cAMP by adjacent sympathetic neurons may promote NGF release from the myocyte, which may then promote neuronal growth and survival). At present, this concept has not been explicitly tested, but provides an interesting hypothesis for communication at the neurocardiac junction (reviewed in Franzoso et al., 2016).

The parasympathetic neuron-cardiomyocyte interface has not been as extensively characterized. Early co-culture studies suggested that embryonic chick cardiomyocytes only developed responses to pharmacological muscarinic stimulation if they were co-cultured with ciliary ganglia, but not when cardiomyocytes were cultured alone (Barnett et al., 1993). Co-culture was also associated with increased expression of inhibitory G-protein subunits (Gi) within the cardiomyocytes, indicating that parasympathetic innervation may have an important role in the development of muscarinic signalling complexes within cardiomyocytes (Barnett et al., 1993). Investigators have recently optimized the differentiation of parasympathetic neurons from human induced pluripotent stem cells (iPSCs) and co-culture of these neurons with cardiomyocytes has been reported (Takayama et al., 2020). These technical advances in iPSC-based technology may pave the way for more detailed mechanistic studies to further define structural and functional connections between parasympathetic neurons and cardiomyocytes.

## Structural/functional organization of cardiac nervous system: reflex control

**Peripheral intrinsic cardiac reflexes.** The cardiac neuronal hierarchy can be viewed as a hierarchy of nested feedback control loops, able to modulate regional cardiac function acutely within a single cardiac cycle as well as chronically (Armour, 2008), as shown in Fig. 1. Less than 15% of the total population of intrinsic local cardiac neurons (LCNs) receive direct preganglionic efferent input (Armour & Hopkins, 1990b; Beaumont et al., 2013b; Gagliardi et al., 1988), with a substantial proportion responding to both sympathetic and/or parasympathetic inputs (Beaumont et al., 2013b; Rajendran et al., 2016) as well as chemical, mechanical and nociceptive stimuli (Rajendran et al., 2016; Salavatian et al., 2016). LCNs therefore play an important role in integrating control of cardiac function between ganglionic plexi and peripheral and central components of the cardiac nervous system

and are therefore potential targets for pharmacological and/or bioelectric therapeutic approaches to modulate dysfunctional reflex responses seen during cardiac disease (Armour, 2008; Beaumont et al., 2015; Gibbons et al., 2012; Hardwick et al., 2015; Salavatian et al., 2022).

**Peripheral intrathoracic, extracardiac reflexes.** Reflex feedback loops governing cardiac function are also dependent on neurons within intrathoracic ganglia (superior and middle cervical and stellate) and mediastinal sympathetic ganglia. These comprise the peripheral intrathoracic, extracardiac reflexes that provide efferent sympathetic control to the intrinsic cardiac ganglia and working myocardium (Armour, 2010; Armour & Ardell, 2004). Intrathoracic extracardiac reflexes receive both excitatory and inhibitory inputs from spinal cord neurons that influence the regulation of regional cardiac electrical and mechanical function (Armour, 1985, 1986a, b; Ardell et al., 2009).

The afferent limb of these reflexes comprises chemo- and mechanosensitive afferent fibres from the heart to its neural control. Chemosensitive TRPV1-expressing fibres with somata located within dorsal root ganglia project to the dorsal root of the spinal cord (Wang et al., 2014; Yoshie et al., 2020). Importantly, these afferent fibres (immunoreactive for calcitonin gene-related peptide (CGRP) and substance P (SP)) traverse the sympathetic ganglia that mediate peripheral intrathoracic extra cardiac reflexes. While coursing through these ganglia, these afferent fibres form varicosities (Happola et al., 1993) that abut neuronal somata presumably influencing neuronal function. Thus, efferent reflex responses from sympathetic neurons within intrathoracic ganglia are activated by both cardiac afferent neurotransmission via dorsal horn of the spinal cord and direct contacts between *en passant* afferent fibres and postganglionic neuronal somata within these ganglia. *In vivo* extracellular recordings demonstrate that the activity of stellate ganglion neurons is closely coupled to the cardiac and respiratory cycles (Gurel et al., 2022; Sudarshan et al., 2021). This suggests that (presumably mechanical) afferent feedback transmitting cardiopulmonary physiology (e.g. cardiac contraction/relaxation and lung inflation/deflation) govern basal function of intrathoracic extracardiac reflex loops.

Recent data using single cell RNA sequencing of sympathetic ganglia (Davis et al., 2020; Sharma et al., 2023; van Weperen et al., 2021) have revealed the diverse cell types that influence the milieu within which postganglionic sympathetic neurons mediate intrathoracic extra-cardiac reflexes. The influence of satellite glial cells and immune cells on pre- to postganglionic neurotransmission as well as excitability and function on postganglionic cardiac sympathetic neurons remains poorly understood. Furthermore, recent data (Sharma et al., 2023) suggest that different sub-populations of

sympathetic neurons (NPY^low/NPY^neg *vs.* NPY^high) mediate basal and enhanced postganglionic sympathetic cardiac reflex responses. This tiered organization of efferent cardiac sympathetic responses may explain differential neurotransmitter/neuropeptide release at low (noradrenaline) and high (noradrenaline with neuropeptide-Y) direct and reflex stimulation of sympathetic ganglia (Kluge et al., 2021; Kluge et al., 2022).

**Central integration.** The central autonomic network (CAN) involves many areas of the central nervous system (CNS) from higher brain areas (cortex, basal ganglia), through midbrain and brainstem structures, down to the spinal cord. The anatomical location of key CAN structures and nuclei are shown in Fig. 2. These areas are important for both sympathetic and parasympathetic control and integration of autonomic changes with behavioural responses, such as the rise in heart rate and blood pressure associated with central command at the initiation of exercise (Krogh & Lindhard, 1913). Whilst the medulla oblongata has long been established as a site for the baroreceptor reflex, this can also be influenced by higher centres (Wilson et al., 1961) such as the cerebellum, inferior olive, hypothalamus and diencephalon areas (Hilton, 1965; Reis & Cuenod, 1962, 1964; Smith & Nathan, 1966).

One of the highest areas (forebrain) shown to be associated with autonomic arousal is the anterior cingulate cortex (ACC) (Gentil et al., 2009), which can suppress baroreceptor sensitivity (Gianaros et al., 2012). More recently, the ACC has been found to be important in central command at the start of exercise (Lamotte et al., 2021), in addition to cognition/ decision making, emotional responses and pain control (Gillies et al., 2019).

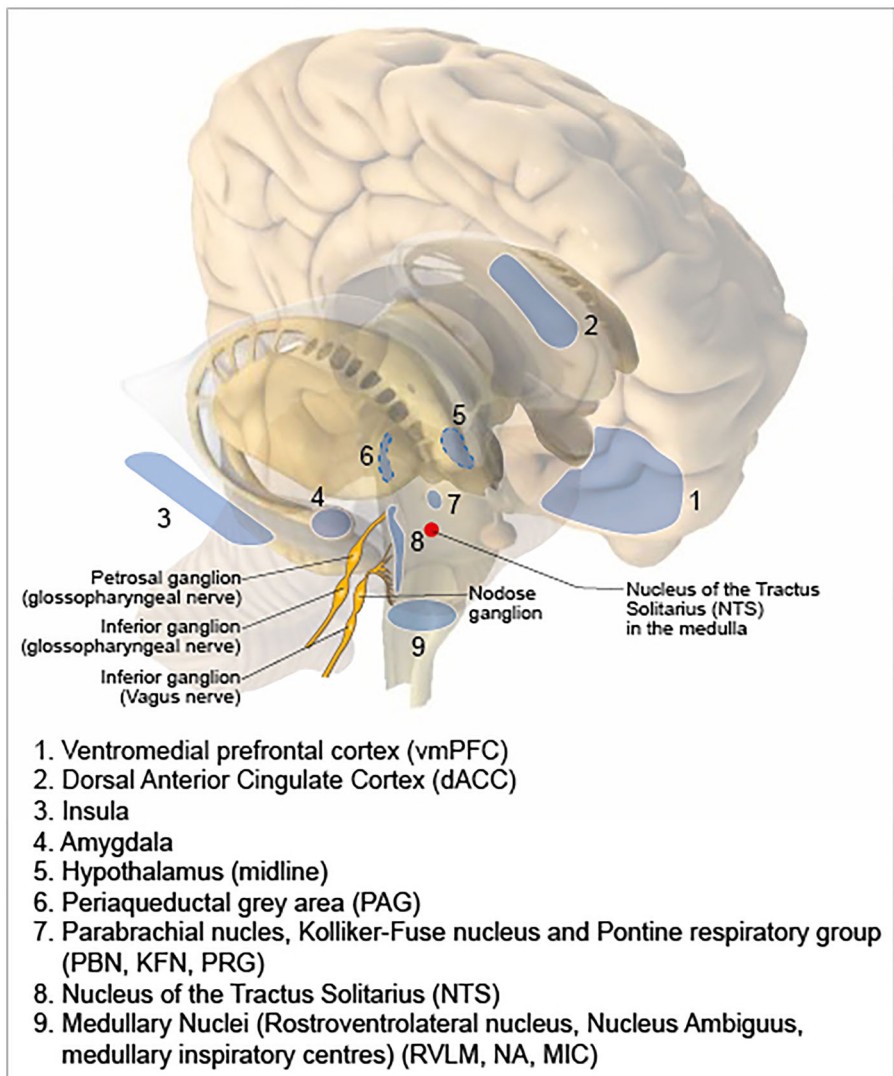

1. Ventromedial prefrontal cortex (vmPFC)
2. Dorsal Anterior Cingulate Cortex (dACC)
3. Insula
4. Amygdala
5. Hypothalamus (midline)
6. Periaqueductal grey area (PAG)
7. Parabrachial nuclei, Kolliker-Fuse nucleus and Pontine respiratory group (PBN, KFN, PRG)
8. Nucleus of the Tractus Solitarius (NTS)
9. Medullary Nuclei (Rostroventrolateral nucleus, Nucleus Ambiguus, medullary inspiratory centres) (RVLM, NA, MIC)

**Figure 2. Key structures of the central autonomic network**
The anatomical location of key structures in the central autonomic network.

On the other hand, the *insula* cortex is important in interoception (the perception of the physiological state of the body) but also has a profound influence on autonomic control. Insula stroke is associated with ECG abnormalities, new onset arrythmias, sudden cardiac death and Takotsubo cardiomyopathy (Abboud et al., 2006; Christensen et al., 2005; Jung et al., 2016; Laowattana et al., 2006), which has led to its putative role as a cause of sudden death in epilepsy (SUDEP) (Li et al., 2017). As with other higher centres, it is likely that the role of insula in autonomic control is related to other functions such as pain processing (that occurs more posteriorly in the insula) (Segerdahl et al., 2015), emotion or cognition as shown through fMRI (Makovac et al., 2018).

Other forebrain areas known to exert top-down autonomic control include the amygdala which is activated during 'fight or flight' reactions (LeDoux et al., 1988) with connections to the hypothalamus and midbrain PAG, parabrachial nucleus, NTS, RVLM and vagal nuclei and can cause sympathetic or parasympathetic activation and secretion of stress hormones (Alheid & Heimer, 1988; Fox & Shackman, 2019; Lamotte et al., 2021). The hypothalamus is also an important part of the 'stress response network' and facilitates integration of autonomic and behavioural responses with endocrine responses via both the sympathetic-adrenomedullary system and the hypothalamic-pituitary-adrenal (HPA) axis (Lamotte et al., 2021; Saper & Lowell, 2014).

The role of diencephalic areas in autonomic control may be largely related to motor control. Electrical stimulation of the midbrain locomotor region (equivalent to the sub-thalamic nucleus in man) drives locomotion and cardio-respiratory responses in decorticate cats, independently of muscle feedback (Eldridge et al., 1981). Studies in humans with implanted subthalamic nuclei electrodes have shown that similar effects occur in patients with Parkinson's disease (Sverrisdottir et al., 2014; Thornton et al., 2002).

Interestingly, the PAG is involved in many functions including descending pain inhibition, fear and anxiety and reproductive behaviour (Kyuhou & Gemba, 1998; Nashold et al., 1969; van der Horst & Holstege, 1998), although stimulation can also alter blood pressure (Kabat et al., 1935). The PAG has reciprocal connections to anterior hypothalamus, thalamus, amygdala, prefrontal cortex and insula (Cameron et al., 1995; Rizvi et al., 1991). Caudally it projects to cardiac vagal preganglionic neurons in the NA, dorsal motor nucleus of the vagus (DMNV) and the NTS (Farkas et al., 1997). Thus, it is likely that the PAG acts as an integrator between higher cortical function and cardiorespiratory responses based on emerging data using deep brain stimulation interventions in humans (Green & Paterson, 2020).

Several areas in the pons deserve special attention. The pedunculopontine nucleus (PPN) is part of the reticular activating system and sends cholinergic projections to the RVLM. In humans, stimulation alters the neural control of the baroreceptor reflex (Hyam et al., 2019). Sitting just caudal to the PPN is the locus coeruleus. This plays a role in stress responses with rich projections to amygdala, prefrontal cortex, PVN, as well as caudal projections to autonomic centres such as RVLM, inter-mediolateral cell column (IML) of the spinal cord and inhibitory projections to parasympathetic nuclei (Lamotte et al., 2021).

**The NTS and modulation of reflexes.** The central site of termination of glossopharyngeal and vagal afferents affecting autonomic efferent activity is the NTS. Afferent fibres enter the medulla at the rostral pole of the NTS and course caudally in a solitary tract. A viscerotopic organization of afferents such that their terminal fields are spatially and anatomically distinct has been proposed. However, many NTS neurones have extensive dendritic morphologies such that the location of their perikarya could be distanced from the terminal fields of their inputs (Deuchars et al., 2000; Paton et al., 2000, 2001). Moreover, many NTS neurones exhibit converging inputs from distinct visceral afferent modalities. It is not possible to know therefore what reflex pathway a neurone participates in when recording or stimulating from distinct NTS regions.

NTS neurons receiving direct afferent inputs as opposed to indirect input via a polysynaptic pathway have been termed 'second order' neurons (Accorsi-Mendonca & Machado, 2013). Whether second order neurons project out of the NTS as well as synapsing on to third order neurons remains unclear. NTS outputs include connections to vagal motoneurones that have multiple functions, premotor sympathetic neurones in the RVLM, oesophageal motoneurones, caudal ventrolateral medullary GABAergic neurones projecting to the RVLM, pre-Bötzinger and Bötzinger respiratory neurones, raphe, pontine and hypothalamic nuclei.

It may be that the organization of visceral afferent termination in the NTS is based on output destination. Thus, distinct afferent inputs that mediate a common reflex response, such as bradycardia, will converge on NTS neurones projecting to cardiac vagal motoneurons. Based on the carotid body, which triggers multiple reflex responses (e.g. cardiovascular, respiratory, hormonal, behavioural), separate transmission lines innervating distinct neurones in the NTS with dedicated outputs were proposed (Zera et al., 2019). This theory may form a basis to explain how, during embryological development, central connectivity is initiated from the end organ, triggering unique retrograde chemotactic signals. Whether separate afferent transmission lines affecting distinct target organ function use unique neuro-transmitters remains to be determined but might permit highly selective pharmacological modulation.

Neurotransmission of visceral inputs to NTS neurones includes a fast glutamatergic mechanism (Aylwin et al., 1997; Neff et al., 1998; Seagard et al., 2003) involving NMDA receptors. This may explain the ability of NTS neurones to show long term facilitation to repeated stimuli present (Yamamoto et al., 2015) that may contribute to the setting of reflex function (e.g. gain, operating ranges). Being the central site of termination of visceral reflex inputs means the NTS presents a powerful site for modulation from other central nervous system sites for adjusting the gain/sensitivity of reflex responses at rest and during changes in state such as sleep/arousal, exercise and 'fight or flight'.

The impressively rich transmitter/modulator substances present in the NTS supports the notion that it is not a simple relay station but a major centre for integration prior to onward transmission to output nuclei in many areas of the brain and spinal cord. For example, the A2 cell group has multiple projections within and outside of the NTS. Using a neuronal lentivirus with a promoter selective for catecholaminergic A2 neurones, a potassium ion channel was over-expressed to induce membrane hyperpolarization (Duale et al., 2007). This resulted in a chronic elevation in arterial pressure during both light and dark phases with no concomitant change in heart rate or baroreflex gain. A similar elevation in arterial pressure could be seen using chemogenetic inhibition of Phox2b in glutaminergic A2 neurones (Melo et al., 2022).

In summary, the NTS plays a major role in the regulation of homeostasis achieved in part through the integration of reflex inputs from multiple peripheral visceral receptors. Additionally, inputs arise from 'signals' in the blood (e.g. hormones) detected by neurones in the adjacent area postrema. The NTS is also an intrinsic chemoreceptive sensing site contributing to hypoxic and hypercapnic ventilatory responses (Fu et al., 2017; Onimaru et al., 2021). The sensitivity to carbon dioxide/pH may be mediated in part by Phox2b containing neurones and adjacent astrocytes (Onimaru et al., 2021).

**Vagal premotor neurons.** Neuronal tracing studies have identified cardiac vagal preganglionic neurones (cVPNs) primarily residing within the brainstem NA and the DVMN (Coote, 2013; Gourine et al., 2016). cVPNs of the NA display rhythmic, respiratory-related patterns of discharge and project B fibre axons. The richest afferent input to the NA comes from the NTS, although it also receives inputs from other areas of the brain. Electrophysiological studies have shown that VPNs that influence heart rate are localized in the ventrolateral part of the NA, although some may be distributed more widely. Two major rhythms determine the patterns of NA neuronal activity: the first is correlated with the arterial pulse and is dependent on inputs from the arterial baroreceptors;

the second is related to the central respiratory rhythm (Gilbey et al., 1984). Powerful baroreceptor modulation and respiratory-related activity distinguishes cVPNs of the NA from the cardiac projecting neurones of the DVMN, which are unaffected by these inputs. The majority of DVMN neurones are tonically active at a low and steady firing rate of 0.5—5 Hz. There is recent evidence that the resting activity of both populations of VPNs in the NA and DVMN increase in response to exercise training (Korsak et al., 2023).

Chronotropic control of the heart is provided predominantly by the neurons of the NA, which synapse on neurons of the intrinsic cardiac ganglia innervating the sinoatrial node, whilst cardiac DVMN neurons control the activity of postganglionic projections that innervate the ventricular myocardium and modulate ventricular excitability and contractility (Machhada et al., 2015, 2016). DVMN activity appears to be critically important for remote ischaemic conditioning (Gourine & Gourine, 2014). Activity of DVMN neurons also has a major impact on myocardial gene expression. Increased DVMN activity causes downregulation of collagen genes and pathways, crucial factors in cardiac fibrosis, adverse remodelling and pathological hypertrophy, highlighted potential mechanisms underlying the beneficial effects of vagus nerve stimulation on the heart (Kellett et al., 2024).

**Sympathetic premotor neurones.** Vasomotor and cardiac sympathetic activities are generated by sympathetic preganglionic neurones of the spinal cord. Excitatory drive from supraspinal regions within the brainstem and the hypothalamus determines the activity of spinal preganglionic neurons. These brain regions include the RVLM, rostral ventromedial and midline medulla, the A5 cell group of the pons and the PVN (Guyenet, 2006). The neuronal circuits of the RVLM appear to be the most important in generating sympathetic tone and provide monosynaptic projections to the spinal sympathetic preganglionic neurones. The RVLM presympathetic circuitry is embedded within the brainstem respiratory network. Therefore, it is not surprising that rhythmic respiratory modulation of sympathetic activity is usually present and can be recorded from the majority of RVLM presympathetic neurones, spinal sympathetic preganglionic neurones and sympathetic nerves.

Results of experimental studies conducted in anaesthetized mice (Schmidt et al., 2018), rats (Marina et al., 2020) conscious sheep (Guild et al., 2018) and humans (Schmidt et al., 2018) had suggested that the brain tissue oxygenation and brain perfusion are the major determinants of the activity of RVLM neurons and therefore of central sympathetic drive. When cerebral perfusion pressure decreases or in response to brainstem hypoxia, activation of the RVLM sympathoexcitatrory neurons triggers compensatory increases in sympathetic

nerve activity, systemic arterial blood pressure and heart rate to maintain cerebral blood flow and brain oxygen delivery, forming a homeostatic feedback loop (Marina et al., 2015, 2020). In ageing, progressive reduction in cerebral blood flow (Christie et al., 2022) may contribute to the developing of systemic arterial hypertension via this mechanism (McBryde et al., 2017).

**Respiratory modulation of autonomic outflow.** During exercise cardiac output and minute ventilation are precisely 'coupled' to optimize efficiency of gas transport to and from tissues, although this coupling can also be observed at rest with respiratory sinus arrhythmia (RSA). Respiration has three phases: inspiration, post-inspiration (first part of expiration) and expiration (or stage II expiration; second part of expiratory pause). One or more phases can modulate autonomic motor outflow (Simms et al., 2009); in humans (Shantsila et al., 2015) via the ventral respiratory group respiratory rhythm/pattern generator located in the ventrolateral medulla. This modulation can change in disease such that both its respiratory phase and/or amplitude can be altered (Fisher et al., 2022) and here we highlight potential mechanisms and its functional importance.

The respiratory rhythmicity of sympathetic activity translates to oscillations in blood pressure first described by Traube and Hering (Hering, 1869; Traube, 1865). So called 'Traube-Hering' waves contribute to the generation and maintenance of vascular tone. Most respiratory modulation occurs during inspiratory and post-inspiratory phases, but in the spontaneously hypertensive rat (SHR) post-inspiratory modulation is enhanced, which may contribute to vasomotor sympathoactivation (Simms et al., 2009). This is associated with increased excitability of post-inspiratory neurones caused by reduced potassium ion channel expression (Moraes et al., 2014). Hypertension induced with chronic intermittent hypoxia generates the appearance of an additional respiratory burst during the expiratory phase (Zoccal et al., 2008), perhaps driven by the carotid body, and may trigger hyper-excitability of the retrotrapezoid nucleus/ventrolateral parafacial nucleus (abdominal pumping; Abdala et al., 2009; Moreira et al., 2024), which couples to the presympathetic neurones of the RVLM (de Britto & Moraes, 2017; Zoccal et al., 2018). Indeed, reinstating respiratory modulation of sympathetic outflow generated greater arterial pressure than when absent (Simms et al., 2009), whereas electrical stimulation of the sympathetic outflow to a hindlimb increased vascular resistance that was greatest when a rhythmic rather than tonic pattern was used (Briant et al., 2015). Modulation of the ventilatory pattern may therefore help alleviate hypertension, which is consistent with finding that slow deep breathing training lowered blood pressure (Joseph et al., 2005).

Activities in both the sympathetic and vagal innervation of the heart are also respiratory modulated (McAllen et al., 2011), not dissimilar qualitatively to vasomotor activity (Boscan et al., 2001). The functional importance of this activity is poorly defined but may contribute to heart rate variability via RSA first recorded by Ludwig in 1847 (Ludwig, 1847). It is mostly blocked by atropine, indicating it is driven by the vagus nerve. Pre- and postganglionic vagal motor neurones show activity predominantly in the post-inspiratory phase (Gilbey et al., 1984; McAllen et al., 2011; Simms et al., 2007), which coincides with heart rate slowing and coupling of medullary post-inspiratory neurones to cardiac vagal preganglionic neurones. Indeed, preganglionic cardiac vagal motoneurones located in the NA exhibited inspiratory-mediated inhibition and post-inspiratory depolarization (Gilbey et al., 1984), suggesting opposing respiratory drives impinging on this cardiac outflow. RSA can be enhanced with exercise training and is suppressed or absent in heart failure, where it is associated with sudden cardiac death (La Rovere et al., 1998).

RSA may be important for ventilation-perfusion matching, equalizing left *vs.* right cardiac output, enhancing coronary blood flow (reviewed recently in Elstad et al., 2018), or even as an energy saving mechanism for the heart (Ben-Tal et al., 2012). Subsequently, a novel pacemaker device has been developed to couple heart rate to respiration and has been tested in both rats (O'Callaghan et al., 2020) and sheep (Shanks et al., 2022) with induced heart failure. After 1 week of pacing, RSA produced a significant improvement in cardiac output, that was sustained for a further 3 weeks of pacing. When the pacemaker was turned off the improvement in cardiac output persisted for several days, suggesting a learned response. RSA pacing reduced NT-proBNP, sympathetic outflow and apnoea incidence and was associated with improvements in regional blood flow and baroreceptor reflex gain, as well as ultrastructural evidence of reverse remodelling (Ben-Tal et al., 2012; Shanks et al., 2022). Whether RSA pacing will provide therapeutic benefit that exceeds that of conventional 'metronomic' pacemaking awaits clinical testing.

### Questions

**Circuit map for normal cardiac control.**

- Can we combine a more detailed anatomical understanding of the diversity of neural populations throughout the autonomic nervous system with better functional understanding?
- Can the molecular basis of afferent and efferent neuronal function be exploited in terms of new therapeutics and biomarkers?

- What are the critical neural elements (peripheral *vs.* central) that translate from animal models to humans?
- Can we exploit patterns and coupling of reflex activity seen in health to help combat disease?
- What are the critical inherent differences in cardiac neural control that predispose individuals to cardiac disease?
- What are the effects of exercise and ageing on integrated neural control of the heart?

### Neuraxial transduction of cardiac pathology

**Overview.** The autonomic nervous system adapts to pathology in the short term to try to maintain adequate cardiac electrical and mechanical function and overall blood pressure and organ perfusion. However, sudden shifts in demand initiated by myocardial ischaemia and long-term remodelling during hypertension and heart failure can be maladaptive and predispose to arrhythmia and/or deterioration in contractile function. It is worth noting that excessive adrenergic activity characterized by high levels of circulating catecholamines and neuropeptide-Y are a negative prognostic indicator post myocardial infarction (MI) (Gibbs et al., 2022; Kleiger et al., 1987; La Rovere et al., 1998) as well as during chronic heart failure (CHF) (Ajijola et al., 2020; Cohn et al., 1984; McDowell et al., 2024; Nolan et al., 1998). In randomized controlled clinical trials, $\beta$-blockers, introduced over 50 years ago, represent the only anti-arrhythmic pharmacological agent known to improve mortality in these conditions (CIBIS-II, 1999; ISIS-1, 1986).

Across all three White Papers in this issue of *Journal of Physiology*, the fundamental premise is that the progression of cardiac disease reflects the dynamic interplay between neurohumoral and cardiac end-effectors. Continual intense neurotransmission from nociceptive afferents, from a site of injury or inflammation, can elicit cellular processes that alter synaptic strength and/or influence neuronal structural connectivity. Pathology can therefore give rise to exaggerated responses of some reflexes while others are blunted, with the potential to develop conflicts between different levels of the neuronal hierarchy. Disease progression, even when initiated locally in the heart, will therefore progress rapidly to the neural–myocyte interface, hormonal systems, peripheral and central mediated reflexes and the interplay with higher centres.

**The relevance of the cardiac neuronal hierarchy in myocardial ischaemia.** During myocardial ischaemia, spinal afferent signalling leads to a marked increase in the central neuronal release of neurotransmitters such as glutamate and substance P which may trigger enhanced excitation of glia (Milligan & Watkins, 2009). Inflammatory mediators including TNF$\alpha$, can also be released from glia during myocardial ischaemia (Niu et al., 2009), which together may cause neuronal sensitization. As well as central reflex arcs, myocardial ischaemia activates peripheral reflexes and increases coherence of activity among neural networks (Armour, 1999; Dale et al., 2020; Malliani & Montano, 2002; Salavatian et al., 2023; Schultz & Ustinova, 1996; Ustinova & Schultz, 1994b). The cardiac nervous system exhibits short-term memory (Ardell et al., 2009; Armour et al., 2002) and longer-term plasticity (Fukuda et al., 2015; Kember et al., 2013a; Wang et al., 2014). During progressive cardiovascular disease, these changes can drive adverse remodelling within the cardiac nervous system, which in turn can exacerbate structural, contractile and electrical remodelling of the heart (Florea & Cohn, 2014; Fukuda et al., 2015; Zucker et al., 1995a) Such neural network interactions are potential targets for neuromodulation to mitigate excessive afferent-mediated sympatho-excitatory reflexes (Hadaya et al., 2023; Herring et al., 2019; Salavatian et al., 2017; Salavatian et al., 2019, 2023).

**The relevance of the cardiac neuronal hierarchy to cardiac arrhythmia induction.** The role of the autonomic nervous system in both atrial and ventricular arrhythmogenesis has received many extensive reviews recently (e.g. Herring et al., 2019; Linz et al., 2019; Sridharan et al., 2022). Here we present a contemporary overview of the general principles, which are explored in more detail in other parts of the three White Papers. Even in patients with severe cardiac pathology, arrhythmias are rare events given the many normal heart beats over a lifetime. When they do occur, arrythmias can be transient and asymptomatic, or sustained and potentially life-threatening. Their initiation is often the 'perfect storm' of an extra stimulus occurring within a vulnerable window that requires a suitable substrate for sustained propagation. The substrate can be static and structural, such as within an established MI scar or around pulmonary veins, or electrophysiological where regional differences exist. This substrate may also be dynamic in that it is influenced by ischaemia, inflammation, haemodynamics, electrolytes, drugs and heart rate. If there is a central area of anatomical or functional conduction block, a zone of slow conduction or unidirectional block can give rise to the formation of a re-entrant circuit. This can sustain ventricular tachycardia (VT) and if fragmentation or wave-break occurs, then multiple wavelets and fibrillation may form (Garfinkel et al., 2000; Weiss et al., 2000).

Each step in this process is probabilistic, with all criteria only rarely being met, although the autonomic nervous system can initiate or modulate every step of this dangerous journey. For example, delayed after-depolarization (DADs), results from cellular calcium over-

load that causes spontaneous diastolic release driving electrogenic Na–$Ca^{2+}$ exchange (NCX) current, which then depolarizes the membrane towards the action potential threshold (Marban et al., 1986). This is promoted by the release of noradrenaline and $\beta$-adrenergic receptor stimulation (Priori et al., 1988), as well release of neuropeptide-Y and Y1 receptor activation (Kalla et al., 2020). Conversely it is opposed by acetylcholine activation of muscarinic receptors and vagal production of nitric oxide (NO) from neuronal nitric oxide synthase (nNOS) in the parasympathetic innervation (Brack et al., 2007; Kalla et al., 2016). Early afterdepolarizations (EADs) are more common when the action potential is prolonged either during extremes of bradycardia, or tachycardia if the action potential duration (APD) does not shorten appropriately and therefore can be modulated by autonomic control of heart rate and action potential duration (APD) (Ben-David & Zipes, 1988).

The shorter the APD and refractory period of an ectopic beat, the more likely it is to be able to produce a re-entrant circuit. The relationship between APD and the preceding diastolic interval is known as the electrical restitution properties (Bass, 1975). $\beta$-Adrenergic receptor stimulation steepens the gradient of electrical restitution to varying degrees from apex to base (Han & Moe, 1964; Mantravadi et al., 2007; Ng et al., 2009), due to differences in regional innervation. This heterogeneity has the potential to be pro-arrhythmic and generate a suitable dynamic substrate for arrhythmia. The increase in heart rate from $\beta$-adrenergic stimulation also can generate calcium driven alternans, which can become regionally discordant, also facilitating re-entry (Tomek et al., 2019). Conversely, the vagi can reduce heart rate and oppose restitution steepening and alternans but is also anti-arrhythmic independently of heart rate as it prevents $Ca^{2+}$ overload and prolongs refractory period.

In the longer term, autonomic remodelling occurs in different pathological states. For example, regional denervation occurs immediately after MI, followed by subsequent reinnervation (Li & Li, 2015). Both generalized and regional denervation (Dajani et al., 2023) and local hyperinnervation from nerve sprouting (Cao et al., 2000) produce heterogeneity in calcium handling and conduction creating dynamic substrate for re-entry. Parallels can be also drawn in persistent AF, where heterogeneous sympathetic innervation and nerve sprouting can occur (Chang et al., 2001; Jayachandran et al., 2000; Wasmund et al., 2003), as well as hypertrophy and heterogeneity of atrial parasympathetic innervation in the ganglionic plexi (Gussak et al., 2019; Yu et al., 2014). As a MI scar heals and CHF develops, changes in other inflammatory and neurohumoral pathways, including a range of growth factors and cytokines, can also become involved (Donahue et al., 2024). This can induce a phenotype switch in neurons from an adrenergic to a cholinergic phenotype. Around 2 weeks after MI, acetylcholine levels in the infarct border zone transiently increase as sympathetic neurons start to express and release acetylcholine and noradrenaline (Kanazawa et al., 2010; Olivas et al., 2016). In CHF, histological and molecular analysis of stellate ganglia demonstrate increased expression of synthetic enzymes of noradrenaline, evidence of inflammation, neurochemical remodelling, oxidative stress and satellite glial cell activation (Ajijola et al., 2017). In a chronic MI model, increased expression of neuronal nitric oxide synthase (nNOS) has been observed in the ventral interventricular ganglionated plexus, dorsal root ganglia and stellate ganglia (Nakamura et al., 2016), whereas decreases are seen in postganglionic cholinergic neurones (Dawson et al., 2008). These changes might be a hallmark of early compensation involving nitric oxide (NO) pathways to modulate sympatho-vagal balance. In the long term, vagus nerve stimulation can also be anti-arrhythmic by reducing inflammation and fibrosis (Calvillo et al., 2011; Wang et al., 2003; Zhang et al., 2009b) and preserving conduction maintaining connexin 43 expression and phosphorylation (Ando et al., 2005; Sabbah, 2011). Moreover, overexpressing of nNOS can restore vagal signalling in states of dysautonomia (Dawson et al., 2008; Heaton et al., 2007). The chronic effects of VNS on myocardial substrate remodelling might have an important role in its antiarrhythmic effect.

**The relevance of the cardiac neuronal hierarchy to heart failure.** In CHF patients, reflexes within the cardiac neuronal hierachy are involved in initating excessive sympatho-excitation. While these changes are initially adaptive with regards to maintaining cardiac output, prolonged sympatho-excitation exacerbates the CHF state. Within the heart itself, myelinated cardiac vagal afferents exhibit a decrease in discharge sensitivity in CHF (Dibner-Dunlap & Thames, 1992; Zucker et al., 1977), due to an intrinsic abnormality of the nerve endings themselves (Zucker et al., 1977, 1979). In the atria, myelinated fibres that end in relatively large diameter and complex sensory endings appear to be markedly deformed (Zucker et al., 1977, 1979). The sensitivity of these unmyelinated fibres is increased in heart failure (Mark, 1987). This may be related to dysfunctional excitability secondary to acid-sensing ion channels (ASICs) (Abboud & Benson, 2015) or abnormalities in Na-K ATPase (Wang, 1994; Wang et al., 1991). Increased levels of cardiac oxidative stress and interstitial products of ischaemia are likely to also be mediators of ion channel dysfunction (Ustinova & Schultz, 1994b).

Cardiac spinal afferents include nociceptors that evoke the sensation of cardiac pain and mediate a sympatho-excitatory reflex via the DRG in response to acute myocardial ischaemia (Brown, 1967; Brown

& Malliani, 1971). Interrupting this reflex in animal models through thoracic dorsal rhizotomy can reduce the incidence of ventricular arrhythmias during myocardial ischaemia (Schwartz et al., 1976). This 'cardiac sympathetic afferent reflex' (CSAR) also appears to have enhanced sensitivity in animal models of chronic heart failure (Wang & Zucker, 1996; Wang et al., 2014, 2017; Zhu et al., 2004), where both afferent discharge sensitivity and the central gain were increased (Ma et al., 1997). Furthermore, the CSAR plays a role in reducing arterial baroreflex gain and enhancing arterial chemoreflex gain (Gao et al., 2004b, 2007), most likely by projections of these afferents to the NTS (Wang et al., 2008). Interrupting the reflex in anesthetized and vagotomized dogs with heart failure using epicardial lidocaine reduced renal sympathetic nerve activity, with little effect in sham animals. In rats with coronary artery ligation-induced heart failure, a similar augmentation of the CSAR was observed (Wang et al., 2014, 2017). The above scenario strongly suggests that cardiac sympathetic afferents contribute, in part, to the sympatho-excitation of heart failure.

TRPV1 is a calcium permeable non-selective cation channel expressed on the peripheral and central terminals of small-diameter sensory neurons. Administration of the ultra-potent TRPV1 agonist resiniferatoxin (RTX) induces calcium cytotoxicity and selective lesioning of the TRPV1-expressing nociceptive primary afferent population (Karai et al., 2004; Sapio et al., 2018). This selective neural ablation has been coined 'molecular neurosurgery' and has the advantage of sparing motor, proprioceptive and other somatosensory functions that are important for coordinated movement and performing activities of daily living. RTX has been used to ablate cardiac spinal afferents by epicardial application in rats at the time of MI (Wang et al., 2014, 2017), showing a reduction in noradrenaline and renal sympathetic nerve activity. This was associated with a reduction in left ventricular (LV) end diastolic pressure, improved diastolic function and reduced cardiac fibrosis even though cardiac systolic function was not significantly improved. Several follow-up studies further demonstrated that epicardial RTX at the time of MI reduced cardiac fibrosis and prevented ventricular arrhythmias in porcine (Yoshie et al., 2020), canine (Zhou et al., 2019a) and rodent (Wu et al., 2020; Zhou et al., 2019b) models. Epicardial or intra-stellate RTX application at the time of MI reduced renal fibrosis, improved renal perfusion, prevented renal dysfunction developing over the subsequent 4 months (Xia et al., 2022) and improved the overall 6-month survival.

Arterial baroreflex sensitivity is depressed in animal models and in humans with CHF (Seravalle et al., 2019; Zucker & Wang, 1991). Baroreflex impairment reduces the ability to supress sympathetic outflow and results in positive feedback to worsen the heart failure state. Reduced baroreflex control of heart rate, has been shown to correlate closely with CHF severity and a poor prognosis in CHF patients with reduced ejection fraction (HFrEF) (Gronda et al., 2014, 2015). Blunted baroreflex control of heart rate also occurs in HF patients with preserved ejection fraction (HFpEF) (Bunsawat et al., 2023; Georgakopoulos et al., 2011; Seravalle et al., 2019). The mechanisms responsible for these baroreflex abnormalities are varied but include both altered sensory transduction at the ion channel level and central remodelling of the baroreflex arc. While changes in Na-K ATPase activity in arterial baroreceptors have been shown in heart failure (Wang et al., 1991, 1992), it is also likely that alterations in both TRP, ASIC and Piezo channels play a role (Abboud & Benson, 2015; Yang et al., 2022; Zeng et al., 2018). Several studies have clearly shown multiple central abnormalities in baroreflex signalling in the heart failure state (Huang et al., 2000; Michelini et al., 2015; Zucker & Gilmore, 1985). Probably, central angiotensin II and reactive oxidant stress play important roles in blunting baroreflex function in heart failure (Gao et al., 2004a; Haack et al., 2013; Tian et al., 2022; Zucker et al., 2014).

A hallmark of CHF is exercise intolerance, which has implications for morbidity, disability and prognosis, and is often the reason a patient seeks medical attention. Stimulation of chemo and stretch receptors in exercising muscle can induce global sympatho-excitation known as the exercise pressor reflex (EPR). In animal models and patients with CHF, this reflex is exaggerated and causes extreme activation of the sympathetic nervous system which may limit muscle blood flow, arteriolar dilatation and capillary recruitment, leading to under-perfusion of working muscle. Furthermore, extreme visceral vasoconstriction by the exaggerated EPR may cause fluid shifts that worsen pulmonary oedema. The exaggerated EPR in animal CHF models may arise from a blunted cardiovascular response to chemical stimulation of group IV afferents, whereas the cardiovascular response to mechanical stimulation of group III afferents is enhanced (Smith et al., 2003; Wang et al., 2010). Overall activation of the EPR tends to produce exaggerated peripheral vasoconstriction with little increase in cardiac contractility or cardiac output (O'Leary, 2006; O'Leary et al., 2004), due to an exaggerated peripheral vasoconstriction with little increase in cardiac contractility or cardiac output. The loss of the cardiac output response of the EPR in heart failure probably stems from an inability to improve ventricular function. Increased mechano-sensitive afferent sensitization in CHF may be driven by muscle cyclooxygenase (COX) signalling, dorsal root ganglia (DRG) purinergic/glutamatergic/brain-derived neurotrophic factor (BDNF) signalling pathways and/or down-regulated voltage-gated potassium ($K_v$) channels in

lumbar DRGs (Antunes-Correa et al., 2014; Hong et al., 2021; Koba et al., 2009; Middlekauff et al., 2004; Middlekauff et al., 2008; Schiller et al., 2019; Tikhonov & Zhorov, 2005; Wang et al., 2010). Recent data also suggest that input from cardiac spinal afferents exaggerates the EPR in animals with heart failure (Mannozzi et al., 2024). In addition to peripheral sensitization of afferents, spinal sensitization mediated by glutamatergic receptors also contributes to the exaggerated EPR in CHF (Wang et al., 2015a). Evidence from muscle afferent recordings indicates that metabo-sensitive muscle afferent (group IV) sensitivity in response capsaicin is blunted in CHF rats (Smith et al., 2003; Wang et al., 2010), suggesting desensitized TRPV1 channels in muscle metabo-sensitive afferents in CHF.

In human studies, similar findings showed that the EPR and mechanoreflex are enhanced in CHF patients (Antunes-Correa et al., 2014; Middlekauff et al., 2000, 2001, 2004, 2008; Momen et al., 2004). Relative to the mechanoreflex, the role of the metaboreflex is controversial, with some studies suggesting an overactive metaboreflex compared with control subjects (Piepoli et al., 1996, 2008), while others suggest that metaboreflex function is blunted (Antunes-Correa et al., 2014; Middlekauff et al., 2000; Sterns et al., 1991). These discrepant conclusions may be due to different measurements of physiological parameters such as ventilation, blood pressure and sympathetic nerve activity.

Multiple studies have shown sensitized peripheral and central arterial chemoreflex function in animals and humans with chronic heart failure (Chua et al., 1997; Chugh et al., 1996; Giannoni et al., 2023; Schultz & Marcus, 2012; Schultz & Sun, 2000; Toledo et al., 2017a). Several preclinical studies have suggested that altered carotid body blood flow, the involvement of the flow sensitive transcription factor KLF2 (Marcus et al., 2018; Schultz et al., 2015; Toledo et al., 2017b) and inhibition of oxygen sensitive potassium channels (Iturriaga et al., 2021) may play a role. Both enhanced chemoreflex activation (even at normal $P_{O_2}$ levels) and the increase in sleep apnoea in patients with heart failure (Cowie et al., 2021; Parati et al., 2016) contribute to sympatho-excitation in heart failure and ventricular arrhymogenesis (Siontis & Somers, 2022). Whilst the use of hyperoxia in heart failure is controversial some evidence suggests that this lowers chemoreflex activation and sympathetic tone (Xing et al., 2014).

## Questions

**Remodelling/adaptions in cardiac control with cardiovascular disease.**

- What are the primary events triggering neural remodelling and how are they stratified based on the underlying pathology?

- How do neurochemical and neuroimmune environments of the ganglionic plexi, DRG, stellate ganglia and supraspinal networks contribute to disease progression?
- How does localized pathology in specific regions of the neural hierachy influence the behaviour of the system as a whole?
- The mechanisms of typical angina observed in male patients *vs.* atypical angina as observed in female patients are relatively unknown. What are the pathophysiological differences between genders and does this extend to other disease states such as hypertension and CHF?
- What is the best approach to control clinical sympathetic storms to prevent serious adverse events?

## Interventional autonomic regulation therapy for cardiac disease

The aim of interventional autonomic regulatory therapy (ART; that involves targeting the cardiac neuronal hierarchy) is to try and stabilize and improve autonomic imbalance, characterized by vagal withdrawal and sustained sympathetic excitation, which plays a significant role in the pathogenesis of cardiac disease. Restoration of such balance could potentially enhance cardiac function, or at least slow disease progression whilst reducing the risk of arrhythmogenesis. Cardiac sympathetic denervation has become part of international guidelines regarding the treatment of ventricular arrythmias in LQT syndrome and CPVT, as is discussed in the clinical White Paper that accompanies this one. However, the translation of other autonomic regulatory therapy into mainstream clinical practice so far has been disappointing. We believe that there are important lessons to be learnt from some of these failures and that there remains significant potential for translation with regard to new sites and ways of interacting with the autonomic nervous and we address some of these below. As understanding deepens and technology develops, this will be a rapidly advancing field. Whilst we briefly mention relevant clinical studies related to different forms of ART, more details are included in the clinical White Paper published alongside this translational White Paper.

**Lessons from neurosurgery.** Deep brain stimulation (DBS), a routine surgical treatment for movement disorders and chronic pain (Boccard et al., 2013; de Jesus et al., 2024) has also been assessed for its effects on autonomic parameters. Similarly, stereo-electroencephalography (SEEG) studies, in which multiple electrodes are inserted temporarily to detect epileptic foci, have found several brain areas that could become putative targets for autonomic control, including

Brodman area 25 (subcallosal neocortex) and insula (Lacuey et al., 2018).

In Parkinson's disease patients, PPN DBS reduces falls in blood pressure associated with standing, with enhanced baroreceptor sensitivity (Hyam et al., 2019). PPN stimulation is the subject of ongoing studies in the context of improving postural drop and other autonomic parameters in patients with Multiple System Atrophy – another neurodegenerative condition affecting the brainstem and causing autonomic failure (Oxford & Trust, 2021). In chronic pain patients undergoing PAG DBS, studies have demonstrated that DBS can be used to increase or decrease blood pressure, depending on whether the electrode is placed in the dorsal or ventral columns, respectively (Green et al., 2005). This led to early speculation that DBS could be used as a therapy for hypertension, albeit with the caveat that the effects were seen in pain patients (Pereira et al., 2010). A case report in a patient with drug and device-resistant hypertension, showed that chronic DBS reduced blood pressure in the long term and that these changes were associated with reductions in sympathetic outflow (O'Callaghan et al., 2017).

### Vagus nerve stimulation (VNS)

*VNS and heart failure.* Direct cervical VNS activates multiple signalling pathways that have the potential to be beneficial in heart failure, including afferent-mediated reflexes (Ardell et al., 2015; Yamakawa et al., 2015a), efferent signalling to cardiac muscarinic M2 and M3 receptors, inhibition of pro-inflammatory cytokines (Janig, 2014; Tracey, 2007) and normalization of nitric oxide signalling (Sabbah, 2011; Sabbah et al., 2011b). For example, VNS increases acetylcholine release from efferent post-ganglionic neurons, which can activate cardiomyocyte M2 muscarinic receptors to induce negative chronotropic, dromotropic and inotropic effects (Levy & Martin, 1979). VNS also has anti-adrenergic effects within the intrinsic cardiac ganglia (Furukawa et al., 1996; McGuirt et al., 1997; Randall et al., 2003), at the neural/myocyte interface (Levy, 1971; Levy & Martin, 1979; Levy et al., 1966) and centrally via afferent feedback (Saku et al., 2014). Overall such changes help improve the balance between energy demands and supply (Buckley et al., 2015; de Ferrari, 2014; Rhee et al., 2015; Sabbah et al., 2011b) and may help myocytes become more stress resistant during heart failure (Beaumont et al., 2015).

VNS also inhibits local cytokine release to prevent tissue injury and cell death (Janig, 2014; Tracey, 2007) via activation of the alpha-7 nicotinic acetylcholine receptor (Wang et al., 2004). including the release from macrophages of high mobility group box 1 (HMGB1) (Wang et al., 2004). Long-term VNS in dogs with CHF reduces plasma HMGB1 levels along with LV tissue TNF$\alpha$

and IL-6 (Sabbah, 2011). VNS also restores nitric oxide signalling. Coronary artery microembolization-induced CHF in canines up-regulates myocardial nNOS and inducible NOS expression (Ruble et al., 2010; Sabbah, 2011), whilst downregulating eNOS (Sabbah, 2011), whilst VNS normalizes the expression of nNOS and improves iNOS and eNOS expression (Ruble et al., 2010; Sabbah, Gupta et al., 2011; Sabbah, Ilsar et al., 2011). Exercise training upregulates nNOS and faciliates vagal responsiveness (Danson & Paterson, 2003), which can be prevented through nNOS gene knockout and rescued with nNOS gene transfer (Danson et al., 2004). Together, these effects serve to augment cardiac energetics and myocardial contractility (Beaumont et al., 2015; Sabbah et al., 2011; Sabbah et al., 2011; Shinlapawittayatorn et al., 2013, 2014; Zhang et al., 2009c). VNS is also associated with an improvement in a range of heart failure biomarkers including plasma levels of BNP, noradrenaline, angiontensin-II and c-reactive protein (Zhang et al., 2009c) and in animal models can confer a marked survival benefit in chronic ischaemic heart disease (Hadaya et al., 2023; Li et al., 2004).

Based on such preclinical evidence, three clinical studies of VNS have reported the effects of VNS in patients with NYHA classes II-IV chronic HFrEF (de Ferrari et al., 2011, 2014; Premchand et al., 2014, 2015; Zannad et al., 2015), all with different approaches to sitmulation of the cervical vagi. These studies are discussed in more detail in the clinical White Paper. When ANTHEM-HF and NECTAR-HF were compared, greater improvements from baseline were observed in ANTHEM-HF in standard deviation in normal-to-normal R-R intervals, left ventricular ejection fraction and Minnesota Living with Heart Failure mean score. When compared with INOVATE-HF, greater improvement in 6-min walk distance was observed in ANTHEM-HF (Anand et al., 2020). The larger ANTHEM-HFrEF Pivotal trial (Konstam et al., 2019) is currently ongoing and yet to report.

The potential for neuromodulation to exert cardio-protection may diminish as the disease progresses and trials to date have targeted the severe end of the heart failure spectrum. It is possible that introducing neuromodulation at earlier stages of the condition could significantly improve its therapeutic efficacy (Dusi et al., 2023; Florea & Cohn, 2014; Fukuda et al., 2015). The stimulus paradigm for VNS needs to be further optimized, patient selection improved, and the potential interactions with the current 'four pillars' of heart failure therapy, namely beta-blockers, angiotensin receptor-neprilysin inhibitors, mineralcorticoid receptor antagonists and sodium-glucose co-transporter 2 inhibitors (SGLT2i) need to be explored, especially given that these are rarely examined together in preclinical models (Anand et al., 2020; Premchand et al., 2019). A 'the stronger the better'

approach may also not be applicable to VNS therapy, given that therapeutic effects can be achieved without significant bradycardia during on-phase stimulation (Ardell et al., 2015; Kember et al., 2014). This point, defined as the neural fulcrum, may be an operating point where bioelectric activation of afferent and efferent projections are balanced such that evoked heart responses are null and disease induced imbalances blunted (Beaumont et al., 2015; Sabbah et al., 2011b).

*VNS and ventricular arrhythmias.* A wealth of pre-clinical and clinical data support the notion that VNS helps prevent both induced and spontaneously occurring ventricular arrhythmias (Billman, 2006; Zipes, 2015). This occurs by both direct (efferent) and reflex (afferent) modulation of the cardiac neural hierarchy (Ardell et al., 2015; Kember et al., 2014; Salavatian et al., 2017; Yamakawa et al., 2015b). Efferent mediated release of ACh activates cardiomyocyte muscarinic receptors (M2 and M3) coupled with antagonism of sympathetic efferent outflows to the heart to reduce heart rate, prolong action potential duration (APD), reduce APD dispersion and affect ventricular restitution/refractoriness (Brack et al., 2007, 2013; Chen et al., 2014; Fukuda et al., 2015; Herring, 2015). Conversely infusion of the muscarinic receptor antagonist atropine increases the occurrence of induced VF (de Ferrari et al., 1991).

While short-term application of VNS mitigates the ventricular arrhythmia potential (Yamaguchi et al., 2018), chronic administration of early-onset reactive right-cervical vagal stimulation exerts substantially greater levels of cardioprotection in the setting of chronic MI (Hadaya et al., 2023). Reactive cervical VNS (cVNS), delivered in a neurophysiological-guided manner (see discusssion on neural fulcrum above), resulted in a substantial improvement in cardiac mechanical function and dramatic reduction in ventricular arrhythmias compared with untreated MI pigs (Hadaya et al., 2023). cVNS stabilized MI-induced electrical heterogeneity in the scar-border zone, reduced anisotropic electrical propagation and conduction block, and normalized myocardial repolarization – key drivers of ventricular arrhythmias *in vivo* (Hadaya et al., 2023). cVNS likewise reduced aberrant structural remodelling of the scar-border zone and principal sympathetic efferent and afferent ganglia (stellate and T1 dorsal root ganglia). In addition, cVNS preserved sympathetic control of the heart after MI. It should be appreciated that substantial reorganization of the cardiac substrate and the cardiac autonomic control occurs shortly after MI and that the earlier the neuromodulation intervention takes place, the better the potential outcome (Dusi et al., 2023). Current FDA approved implantable VNS interventions have temperal limitations owing to off-target effects. Future studies should consider refinements in electrode

interfaces and stimulation protocols that could allow for selective targeting of cardiac efferent fascicles thereby mitigating off-target effects such as cough and gastro-intestinal discomfort (Ravagli et al., 2023). Tragus nerve stimulation, which modulates vagal afferent input to impact autonomic outflow, is also a non-invasive option for early neuromodulation that has had success with regard to small clinical studies in atrial fibrillation, as discussed below and in the clinical White Paper. It too could have clinical utility in the context of heart failure, both in its own right and/or as a way of determining patient selection for chronic cVNS implants (Stavrakis et al., 2020a).

*VNS and atrial arrhythmias.* Vagal stimulation can both increase or decrease the susceptibility to atrial arrhythmias (Chen et al., 2014; Lee et al., 2013; Nadeau et al., 2007). Higher intensity stimulation tends to increase atrial fibrillation (AF) inducibility (Zhang et al., 2009a, d), whilst lower intensity stimulation can stabilize atrial electrical function (Chinda et al., 2015; Stavrakis et al., 2015) with minimal adverse effects (Sheng et al., 2011; Zhang & Mazgalev, 2011; Zhang et al., 2009a). Low-level VNS therapy suppresses AF and stellate ganglion hyper-activity in ambulatory canines (Chinda et al., 2015; Shen et al., 2011) and more recent clinical studies have shown chronic intermittent low-level tragus stimulation to lower AF burden and TNFα levels compared to sham control stimulation, in patients with paroxysmal AF (Stavrakis et al., 2020b). The anti-inflammatory effect of vagus nerve stimulation may be particularly important in both AF and heart failure with preserved ejection fraction (Bazoukis et al., 2023) where improved patient symptoms have been observed in small clinical studies (Kumar et al., 2023).

Recent studies have evaluated specific targets within the intrinsic cardiac nervous system that are impacted by neuromodulation interventions (Armour et al., 2005; Gibbons et al., 2012; Richer et al., 2008). Specifically, mediastinal nerve stimulation (MNS) reproducibly evokes AF by excessive and heterogeneous activation of intrinsic cardiac neurons (Gibbons et al., 2012). Using extracellular recordings and responses to physiological stressors, intrinsic cardiac neurons were classified as afferent, efferent, or convergent (afferent and efferent inputs) (Salavatian et al., 2016). The capacity of MNS to modify IC activity in the induction of AF was determined before and after preemptive cervical vagal stimulation. Convergent LCNs were preferentially activated by MNS (Salavatian et al., 2016). Preemptive cervical VNS reduced MNS-induced changes in LCN activity while mitigating MNS-induced AF (Salavatian et al., 2016). These anti-arrhythmic effects persisted post-VNS for, on average, 26 min (e.g. VNS exhibits memory) (Salavatian et al., 2016). Future studies should define the primary neuro-transmitters and neuropeptides released by MSN and

how VNS mitigates the intrinsic cardiac neural network instability that leads to atrial electrical instability.

These data indicate that VNS therapy can favourably modify the underlying pathophysiology of heart failure, atrial fibrillation and malignant ventricular arrhythmias in animal models. These benefits are the result of targeting multiple components of the cardiac neuro-axis to: (1) restore 'central parasympathetic drive, (2) mitigate adverse remodelling of sympathetic inputs to the heart, (3) suppress pro-inflammatory cytokines; (4) normalize nitric oxide signalling pathways and (5) alter myocyte energetics. Future studies on the efficacy of VNS for cardiac therapeutics need to optimize stimulation parameters, improve patient selection and identify the best timing of the intervention during disease progression. Key similarities and differences between direct cervical stimulation compared to low level tragus stimulation also need to explored in more detail. As VNS engages multiple levels of autonomic control (Ardell et al., 2015; Yamakawa et al., 2015b), future preclinical studies should be designed to employ the entire cardiac nervous system, examine long-term therapeutic benefits and off-target side effects, as well as evaluate the therapy in the context of current clinical heart failure therapy.

*Spinal cord stimulation.* Spinal cord stimulation (SCS) was first proposed over 20 years ago for the treatment of refractory angina pectoris (Foreman & Linderoth, 2012; Mannheimer et al., 2002; Zhang et al., 2014) and has since been identified as being potentially anti-arrhythmic (Cardinal et al., 2006; Gibbons et al., 2012; Lopshire et al., 2009). SCS also alters peripheral ganglia neural processing (Ardell et al., 2009; Armour et al., 2002; Gibbons et al., 2012) and signalling at the neural-end organ interface (Cardinal et al., 2006; Southerland et al., 2007). Overall it appears to minimize apoptotic changes (Southerland et al., 2007; Southerland et al., 2012) and preserve contractile function in preclinical models (Lopshire & Zipes, 2012; Lopshire et al., 2009). However, to date it has not become part of mainstream clinical therapy.

*Preclinical studies.* Cardiac sympathetic afferent visceral inputs synapse at lamina I, V, VII and X of the C8-T9 dorsal horn (Armour & Kember, 2004; Foreman, 1999; Foreman & Linderoth, 2012). Peripheral and central reflex processing of this afferent signalling contributes to sympatho-excitation that enhances the progression of heart failure (Florea & Cohn, 2014; Zucker et al., 2012) and the potential for ventricular arrhythmias (Fukuda et al., 2015). SCS aims to dampen such sympathetic reflex responses for example, via the release of neuro-modulators such as dynorphin that blunt the release of primary afferent related neurotransmitters (such as substance P) and alter basal activity of sympathetic pre-ganglionic neurons (Ding et al., 2008a, b). In disease models of myocardial ischaemia, spinal modulation

reduces cardiac arrythmia by suppressing hyperactivity and neural synchrony in the IML (Salavatian et al., 2023). Both SCS and cardiac sympathetic denervation attenuate reflex-mediated noradrenaline release during ischaemia thus preventing ventricular fibrillation (Ardell et al., 2019). Likewise, within the intrinsic cardiac nervous system, reflex responses to transient ischaemia are blunted by SCS (Armour et al., 2002; Foreman et al., 2000); lasting for up to an hour after SCS (Armour et al., 2002). This neural memory could allow intermittent SCS to be employed for cardiac control and angina management (Armour, 2008; Foreman & Linderoth, 2012; Kember et al., 2013b). Local circuit neurons also appear to be an indirect primary target of SCS therapy (Ardell et al., 2009; Gibbons et al., 2012), where SCS modifies synaptic function without directly altering transmembrane properties (Ardell et al., 2014). As a consequence, SCS reduced their electro-physiological activity in an animal model of chronic ischaemia (Cardinal et al., 2004) and in a canine HF model reduced susceptibility to ventricular arrhythmias, whilst improving left ventricular function (Lopshire & Zipes, 2012; Lopshire et al., 2009).

The level of SCS is also likely to be important, with high thoracic SCS effectively targeting thoracic elements of autonomic control (Ardell et al., 2009; Foreman et al., 2000; Lopshire & Zipes, 2014; Southerland et al., 2007), whilst high cervical SCS has the potential to target both thoracic and visceral elements (Foreman & Linderoth, 2012; Southerland et al., 2012). In preclinical studies, the efficacy of SCS is optimum when applied preemptively (Foreman et al., 2000; Southerland et al., 2007), although cardioprotective effects have been observed when applied in chronic HF following MI (Ardell et al., 2014; Lopshire et al., 2009).

*SCS clinical studies.* The Defeat-HF trial (NCT01112579) was a randomized, multicentre, single blind study of 66 patients with systolic CHF, where SCS was cycled 12 h on and 12 h off. Unfortunately, the study failed to show improvement in left ventricular function (Wang et al., 2015b; Zipes et al., 2015). In contrast, the Spinal Cord Stimulation Heart study, a multicentre, prospective, pilot trial involving SCS in patients with systolic CHF (EF 20%–30% and NHYA class III), reported that continuous T1-T3 SCS (50 Hz for 24 h a day) improved NYHA classification, quality of life, left ventricular end systolic volume and peak oxygen consumption (Tse et al., 2015). Temporary SCS has been applied to coronary artery bypass graft patients in a randomized controlled clinical trial in which a percutaneous lead was placed preoperatively and stimulated for 30 days (Romanov et al., 2022). In this 53-patient trial, SCS reduced atrial fibrillation burden from 30.7% to 3.8%. It is evident that additional mechanistic studies are required to understand

the mechanisms whereby SCS exerts its effects on (1) central *vs.* (2) peripheral components of the cardiac neuro-axis (Ardell, 2016). It is also possible that non-invasive transcutaneous electrical or magnetic stimulation of the spine, developed to treat spinal cord injury where orthostatic hypotension can be life-threatening, may provide a better approach (Mikhaylov et al., 2020; Phillips et al., 2018), but further research is warranted.

The application of SCS to treat orthostatic intolerance or hypertension is more appealing than using DBS because of the much lower risk-benefit ratio. A recent review by Laws et al describes 59 studies looking at SCS and the effects on blood pressure and heart rate (Law et al., 2024). In general, there were numerous studies demonstrating an increasing blood pressure with cervical, high thoracic and mid-low thoracolumbar SCS and some studies showing that cervical/mid-to-low thoracolumbar SCS can have the opposite effect. Whilst the results seem to be equivocal, both in effect size and the direction of change, SCS has been used clinically in the context of spinal cord injury and in multisystem atrophy. In these patients, SCS consistently increases resting blood pressure and improves symptoms of orthostatic intolerance or dysreflexia (Aslan et al., 2018; Harkema et al., 2018; Legg Ditterline et al., 2021; Squair et al., 2021, 2022). It appears that stimulation of the IML can be used to increase sympathetic outflow in individuals with a reduction or dysregulated autonomic outflow but its effects on hypertension are equivocal as most studies involve chronic pain patients, where symptoms of pain were also improved by SCS (Holwerda et al., 2021). Moreover, emerging evidence in humans has demonstrated clinical utility for dorsal root ganglion stimulation reducing sympathetic outflow and long-term blood pressure (Sverrisdottir et al., 2020).

*Carotid body (CB) ablation.*    In many cardiovascular diseases, excessive sympathetic activation can be driven at least in part by sensitization of the peripheral chemo-receptor reflex and aberrant discharge generation from the CB that drives sympathetic activity (Pijacka et al., 2016) and evokes hypopneas/apnoeas as seen in heart failure (Lataro et al., 2023). This has led to a series of interventional studies to remove the CBs in animals and then human patients with cardiovascular disease. This was not unprecedented, as the first CB ablation occurred in Japan for the successful treatment of asthma related dyspnoea in humans (Nakayama, 1961). An extensive review of the available literature deemed that the procedure was safe and without serious adverse effects (Paton et al., 2013). An advantage of CB ablation is that pathological signalling can be detected, allowing identification of patients who may benefit. There are two practical and non-invasive ways to assess CB hyperreflexia and discharge generation. First, brief exposure to hypoxia evokes a hypoxic ventilatory response (HVR) and,

based on the relationship between minute ventilation and blood oxygen de-saturation, the sensitivity is calculated. The greater the sensitivity, the greater disease severity. In heart failure patients, a high HVR was prognostic for poor survival (Ponikowski et al., 2001). Second, transient suppression of CB with hyperoxia or low dose dopamine infusion suppresses ventilation (Pijacka et al., 2016) and muscle sympathetic nerve activity (Sinski et al., 2012) and reveals increased CB discharge generation.

Given this dysfunctional activity of CBs, proof of principle studies were designed to test causation in cardiovascular diseases. First, in heart failure, bilateral CB ablation improved cardiac pump function, lowered NT pro-BNP levels, reduced sympathetic activity and prevented central sleep apnoeas in rats with heart failure (Marcus et al., 2014). In human patients, muscle sympathetic nerve activity decreased, the $\dot{V}_E/\dot{V}_{CO_2}$ slope improved but left ventricle systolic function was unchanged. However, a worsening of oxygen de-saturation at night was found in 40% of CB resected (mainly bilateral) patients, raising concern. Unlike the rats, which exhibited central sleep apnoea, most of the human heart failure patients expressed obstructive sleep apnoeas, which were not prevented by CB ablation. Second, in spontaneously hypertensive rats, CBs exhibited hypertonicity insensitive to hyperoxia (Pijacka et al. 2016) and bilateral ablation lowered both arterial pressure (∼17 mmHg) and sympathetic nerve activity; it also improved baroreceptor reflex gain and reduced systemic inflammation (McBryde et al., 2013). The human trial in drug-resistant hypertensive participants removed a single CB only and resulted in a substantial drop of >20 mmHg in 24 h ambulatory blood pressure and a reduction in muscle sympathetic nerve activity that was maintained until the end of the assessment period (6 months; (Narkiewicz et al., 2016)). This effect was not consistent in all patients and a retrospective analysis indicated that those patients that responded best had the highest HVR and displayed a greater depression of minute ventilation during low dose dopamine infusion (Pijacka et al., 2016).

Whilst there is the ability to pre-screen and select patients with CB dysfunction (hypoxia and/or hyperoxia), its removal clearly compromises both ventilation at night and the ability to terminate apnoeic episodes. With the idea that the CB may use separate sensory pathways for regulating its different reflex components, the recent data in spontaneously hypertensive rats demonstrating that P2X3 receptor antagonism attenuated both CB evoked and basal sympathetic nerve activity without affecting ventilatory control (Pijacka et al., 2016; Zera et al., 2019) provides justification for testing such drugs for the future treatment of resistant hypertension and or heart failure in humans.

*Carotid sinus stimulation*

*Preclinical studies.* Any attempt at harnessing arterial baroreceptors to provide therapeutic control of arterial blood pressure and sympathetic activity must logically start with determining that they control blood pressure chronically. The following section reviews the literature for and against this. In the landmark study of Cowley et al. (1973) they state: "the primary function of the baroreceptor reflex is not to set the chronic level of arterial blood pressure but, instead, to minimize variations in systemic arterial blood pressure" (Cowley et al., 1973). This became the dogma and has appeared in many text books. In the 1980s, Krieger and colleagues working with rats showed that sinoaortic denervation caused chronic increases in mean arterial pressure (Trindade & Krieger, 1984) and renal sympathetic nerve activity (Irigoyen et al., 1988), suggesting a long-term role for baroreceptors in regulating arterial pressure. An aspect of cutting nerves and removing a major afferent signal to the brain could lead to central plasticity, making interpretation of sinoaortic denervation data equivocal. Thus, Thrasher (Thrasher, 2002) sectioned both aortic depressor nerves and one carotid sinus nerve but left the other carotid sinus nerve intact. He unloaded the neutrally innervated carotid sinus by placing a blood flow restrictor on the common caroitd artery in conscious dogs. This led to 'chronic' hypertension (+22 mmHg) that persisted until the restriction was removed 1 week later. It was concluded that baroreceptors can provide long-term control of arterial pressure. The study was repeated for 5 weeks and although the magnitude of the rise in arterial pressure waned over time it remained modestly higher than controls, supporting the prior conclusion that baroreceptors can control arterial pressure chronically (Thrasher, 2005).

The first studies in animals where a carotid sinus nerve was stimulated electrically were performed in anaesthetized dogs and were found to lower blood pressure in normotensive (Warner, 1958) and hypertensive conditions (Griffith & Schwartz, 1964); the latter authors demonstrated that the depressor response was greatest on resecting the unstimulated carotid sinus nerve that compensated for the fall in blood pressure. It should be noted that the carotid sinus nerve is not restricted to baroreceptor afferents but also contains many carotid body fibres which have been shown to be coactivated (Katayama et al., 2015); selection of stimulation parameters may help to preferentially drive baroreceptors. Knowing how long the depressor effect might last for and whether the nerve remains viable with continued stimulation are necessary precursors for translation to humans. Neistadt and Schwartz (Neistadt & Schwartz, 1967) reported that reductions in arterial pressure were maintained for up to 6 h and the carotid sinus nerve remained viable for 6 months if stimulation was intermittent. In conscious normotensive rats, continuous stimulation of the aortic depressor nerve was found to be more effective than intermittent stimulation over 1 h (Brognara et al., 2016) and also suppressed release of inflammatory mediators (Ribeiro et al., 2020). The work of Lohmeier and colleagues demonstrated the longevity of response over 3 weeks using continuous stimulation in conscious dogs; the falls in blood pressure (∼20 mmHg) and sympathetic activity were not compensated by either central nervous system habituation or from changes in adrenergic reactivity of the vasculature (Lohmeier et al., 2010). Moreover, it was found that carotid sinus stimulation remained effective after renal nerve denervation (Lohmeier et al., 2007), suggesting distinct mechanisms of action and the potential option that a patient not responding to renal denervation might respond to baroreceptor stimulation. A point of difference was that Lohmeier et al. (2010) stimulated the carotid sinus directly and not the carotid sinus nerve, which overcame the potential of co-activation of the carotid bodies. All told, such preclinical data provided evidence that from both technical and biological standpoints baroreceptor stimulation can be considered an effective anti-hypertensive therapy that reduces blood pressure by mechanisms including withdrawal of sympathetic vasomotor activity (Durand et al., 2009).

*Clinical studies.* The study by Carlsten et al. (Carlsten et al., 1958) was one of the first to show that acute electrical stimulation of the carotid sinus nerve reduced blood pressure and heart rate in humans who were undergoing neck surgery. Then followed a series of reports in the 1960s and 1970s describing how electrical stimulation of the carotid sinus nerves lowered blood pressure in refractory hypertensives and treated both intractable angina and supraventricular tachycardia (Bilgutay & Lillehei, 1966; Braunwald & Sobel, 1969; Braunwald et al., 1967; Epstein et al., 1969; Reich et al., 1967; Schwartz et al., 1967). These studies provided proof of principle but longevity of effect was limited and patient numbers were often small. There were also adverse effects (coughing, gagging, dyspnoea, dysphonia, dysphagia, hyperpnoea, hyperesthesias, or paresthesias), surgical complications due to the bulky nature of the devices, damage to the carotid sinus nerve and limited battery life. Limitations to altering stimulation parameters once the devices were implanted were also common, which was problematic because parameters set under anaesthesia were often inappropriate in the conscious patient. These technical issues needed to be overcome before it was practical to reconsider stimulating baroreceptors for therapeutic gain in human patients.

The first device in the modern era was named 'Rheos.' This was a relatively large glove shaped fabric

electrode designed to be wrapped around the carotid sinus bilaterally. It did not need to be accurately placed on the baroreceptor afferent terminal field and voltage/frequency could be altered post-operatively. In drug resistant hypertensive patients, an early study showed repeatable lowering of arterial pressure associated with inhibition of both muscle sympathetic nerve activity and plasma renin during ∼6 min stimulation periods (Heusser et al., 2010). Subsequent studies with the Rheos device reported efficacy of blood pressure lowering at 12 months (Alnima et al., 2012) and at 6-year follow up (de Leeuw et al., 2017). It was soon established that unilateral stimulation on the right side was more effective at blood pressure lowering compared to bilateral stimulation (de Leeuw et al., 2015).

As a result of these findings the next generation device 'NeoStim' (CVRX) was produced, which comprised a small button shaped electrode that required suturing to the carotid sinus. This device was less bulky and easier to implant but the limited contact surface area of the electrode necessitated accurate placement on the baroreceptor afferent terminals, meaning repeated testing during surgery for the optimal position. A caveat was that in some cases the location of stimulation giving a good baroreflex response under general anaesthesia did not remain efficacious in the awake state. In such cases increasing the stimulation intensity was needed to see a fall in arterial pressure but this resulted in intolerable side effects due to stimulus spread (Heusser et al., 2016). These issues and other technicalities were discussed, offering guidance for management for these patients (Koziolek et al., 2017). However, safety studies indicated that multi-month stimulation showed no dissolution of the platinum or iridium coatings from the electrode and there was no tissue damage to either the tissue stimulated or surrounding areas and no evidence of increased inflammation or arterial stenosis after 6 months of continuous stimulation (Wilks et al., 2017).

Studies using the NeoStim device reported similar magnitude falls in blood pressure as seen with the Rheos device accompanied by small falls in heart rate (∼5–10 beats) and improvement in renal function (Wallbach et al., 2018). Thus, device mediated baroreceptor stimulation appears effective in lowering blood pressure in drug resistant hypertensive patients through inhibition of sympathetic nerve activity. The disproportionate nature of the magnitude in the fall in heart rate is curious and remains to be explained but may be antagonized by the remaining non-stimulated baroreflex regions.

Barorceptor activation therapy has also been tested as a treatment for heart failure. In a study of NYHA class III heart failure patients, significant improvements were reported in ejection fraction, NT-proBNP, quality of life and exercise tolerance (Zile et al., 2015). A single patient case study also reported reductions in atrial fibrillation and ventricular arrythmias that were associated with improvement in systolic function (Wang et al., 2023). Moreover, cost analysis indicated treatment savings within 2 years of initiating baroreceptor stimulation (Bisognano et al., 2021). The use of baroreceptor stimulation in heart failure appears promising but requires further validation.

Given the promising outcome of studies using electrical stimulation of baroreceptors, an endovascular stent was engineered ('Mobius') to sit within and auto-expand the carotid sinus; this was designed to both excite and sensitize baroreceptor stretch sensitive afferent terminals of humans with resistant hypertension. Early trials with Mobius produced substantial falls in ambulatory arterial pressure (∼20 mmHg), with small reductions in heart rate but no significant reductions in muscle sympathetic nerve activity (van Kleef et al., 2021). Falls in blood pressure have persisted at the 3-year follow up (van Kleef et al., 2022). This device has also been used in patients with heart failure (NYHA group III), with positive outcomes in 17 patients at the 12-month follow-up including improved quality of life, reduced NT-proBNP, improved ejection fraction and exercise tolerance (Piayda et al., 2022).

In conclusion, baroreceptor modulation therapy, whether electrical or via an endovascular expansion stent, appears promising in reducing blood pressure and suppressing symptoms in heart failure by reducing heightened levels of sympathetic activity. Critics would also vouch that the technique is relatively invasive and in some patients threshold stimulation intensities required to lower blood pressure cause off target adverse effects that are not tolerable. Moreover, stimulation is always 'on' and without feedback, which is unphysiological. Given that some patients may have naturally occurring night time dips in blood pressure, continuous stimulation may cause issues with hypotension. Thus, future blood pressure controlling devices would benefit from being modulated by physiological feedback of arterial pressure.

### Renal denervation

*Preclinical studies.* Decades of preclinical studies have demonstrated that total renal denervation (TRDN; efferent + afferent) modulates sympathetic nerve activity to several organs including the heart (Osborn & Foss, 2017). Although TRDN targets both efferent and afferent renal nerves, the beneficial cardiac effects in various disease models are assumed to be due to interruption of afferent renal nerves specifically. In relation to outcomes related to cardiac function, TRDN reverses the deleterious effects of MI induced heart failure on left ventricular function (Nozawa et al., 2002), ventricular sympathetic innervation (Pinkham et al., 2017) and post MI cardiac remodelling (Hu et al., 2012) in the rat. TRDN also has been shown to reduce atrial remodelling in hypertensive rats with metabolic syndrome and this was associated with a reduction in proinflammatory and fibrotic signalling

(Selejan et al., 2022). In dogs, TRDN reduces ventricular substrate remodelling in pacing induced heart failure (Dai et al., 2014). TRDN has also been shown to reduce electrical remodelling in the heart, resulting in a reducing inducibility of ventricular fibrillation (Guo et al., 2014). The inducibility of ventricular fibrillation is increased by both obesity and heart failure in a rabbit model and these are both prevented by TRDN (Yamada et al., 2020). In an ovine model of pacing induced heart failure, catheter based TRDN was associated with a decrease in heart rate as a result of increased cardiac vagal tone and a reduction in cardiac sympathetic tone (Singh et al., 2017).

A method to specifically ablate afferent renal nerves in preclinical models has been developed, thus enabling a targeted approach to study of the role of afferent renal nerves in the modulation of sympathetic nervous system activity under physiological and pathophysiological conditions (Foss et al., 2015). Afferent renal denervation (ARDN) is achieved by periaxonal application of the TRPV1 agonist capsaicin, which results in calcium induced cell toxicity in afferent renal nerves that express this channel, while sparing efferent renal nerves that do not. This method of ARDN attenuates chronic activation of the sympathetic nervous system in two models of hypertension in both rats and mice, the DOCA-salt model (Banek et al., 2016, 2018, 2019; Baumann et al., 2022) and the two kidney–one clip (2K-1C) model (Lauar et al., 2023; Ong et al., 2019), as well a model of chronic kidney disease induced hypertension (Veiga et al., 2020). In the rat DOCA-salt model, this is consistent with direct measurement of the basal level of afferent renal nerve activity which is 2.5-fold higher in DOCA-salt hypertensive rats than normotensive controls (Banek et al., 2016).

*Clinical studies.*   Catheter based renal nerve denervation (CBRDN) has emerged as a potential treatment of hypertension in humans. Initial enthusiasm for CBRDN was temporarily stalled following the failure of the Simplicity HTN-3 trial, which subsequently was found to be due to several issues related to the clinical trial design and execution (Osborn & Banek, 2018). Confidence in this therapy has been restored by recent clinical trial results from three different companies employing three different modalities: radiofrequency-based ablation (Medtronic), ultrasound-based ablation (ReCor Medical) and chemical based ablation using alcohol (Ablative Solutions) (Rey-García & Townsend, 2022). Similar results have been obtained by all three approaches with long-term decreases in arterial pressure like that achieved with a single pharmacological therapy (Rey-García & Townsend, 2022).

Following the earlier clinical trials, there were case studies suggesting that CBRDN reduced the frequency of lethal ventricular arrhythmias in the setting of LV dysfunction (Bradfield et al., 2014; Remo et al., 2014) and provided benefits in preserved ejection fraction, thereby resulting in less structural remodelling (Brandt et al., 2012). There are now numerous follow-up studies from the CBRDN clinical trials that are consistent with the idea that CBRDN chronically reduces activity of the sympathetic nervous system to several organs including the heart, which results in additional beneficial effects beyond reducing arterial pressure. A study using cardiac magnetic resonance imaging showed that stroke index, cardiac index and stroke work index were all reduced by CBRDN compared to sham in patients with hypertension (Lurz et al., 2020). Both left ventricular function and exercise capacity were increased following CBRDN in patients with heart failure with reduced ejection fraction (Fukuta et al., 2022; Li et al., 2023). CBRDN in combination with pulmonary vein isolation (PVI) resulted in greater improvements in arterial pressure and estimated glomerular filtration rate than PVI alone, thereby reducing the risk of atrial fibrillation (Nawar et al., 2023). CBRDN has also been reported to reduce the incidence of ventricular arrythmias and implantable cardioverter defibrillator shocks (Garg et al., 2021; Prado et al., 2021).

*Irreversible and reversible decentralization of autonomic ganglia.*   One approach to address excessive sympathetic drive in the context of VT storm is through cardiac sympathetic denervation. Stellate ganglion block and thoracic epidural anaesthesia have emerged as reversible emergency procedures in the management of recurrent VT as a bridge to catheter ablation and/or surgical cardiac sympathetic denervation (Malik & Shivkumar, 2024). Surgical intervention then includes the removal of both stellate ganglia as well as their associated paravertebral chains (T1/2-T4; Cauti et al., 2025; Lee et al., 2022), often via a video assisted thorascopic (VATS) approach. A more recent technique (Buckley et al., 2016) describes a less extensive resection of T1-T2. This approach was proven to meet the therapeutic goals while minimizing damage to sensory and sympathetic motor control of upper limb, neck and thoracic wall that occurs with a full T1/2-T4 resection. Surgical cardiac sympathetic denervation is used routinely in specialist centres to treat recurrent ventricular arrythmias associated with structural heart disease, as well as inherited conditions such as long QT syndrome type 1 (LQT1) and catecholaminergic polymorphic ventricular tachycardia (CPVT). This is discussed in further detail in the third White Paper published alongside this one. Despite these advances, this approach is irreversible and can have off target effects such as hyperalgesia and hyperhidrosis and well as complications related to the surgical access.

Although there are many ways to instigate ART, there are trade-offs between different interventions.

Ablations and surgery are destructive and irreversible. Local anaesthetics are broad acting and not immediately reversible and current pharmaceuticals have significant side effects. Electrical nerve block is a localized treatment that can be applied instantly, is rapidly reversible and can be titrated to customizable dosages. Kilohertz frequency alternating current (KHFAC) has been used to provide a dynamic depolarization of the targeted nerve, which prevents the propagation of action potentials by maintaining voltage-gated sodium channels in a persistent state of inactivation, thus providing a depolarization block of excitation. This technique was originally proven on motor nerves (Bhadra & Kilgore, 2004b, 2005; Kilgore & Bhadra, 2006), but has been more recently proven in the autonomic nervous system as well (Buckley et al., 2017). KHFAC, however, produces spurious activation when initiated, referred to as 'onset activity' (Bhadra & Kilgore, 2005). This onset activity cannot be mitigated by a ramping up of the waveform amplitude (Miles et al., 2007) and can produce a prolonged effect depending on the targeted nerve.

KHFAC neuromodulation can be applied to T1-T2 on the paravertebral sympathetic chain to downregulate sympathetic tone (Buckley et al., 2017). The amount of nerve block was evaluated by placing a stimulating electrode at T3 and observing changes in heart rate, LV d$p$/d$t$ and LV Pressure (LVP). KHFAC was applied prior to T3 stimulation to measure the onset activity. The onset activity results in a pronounced tachycardia for ~100 s before the block takes effect. Once the nerve has been blocked, the stimulation at T3 no longer produces a change in the haemodynamic parameters. Following the cessation of KHFAC, the stimulation was reapplied at T3 and the change in the haemodynamic parameters was completely restored, demonstrating the reversibility of electrical nerve block. To demonstrate the localizable effect of KHFAC, a second stimulation electrode was placed at T1. Stimulation at T1 produced a clear change in the haemodynamic parameters prior to KHFAC application. During KHFAC application, the same effect was observed, showing that the KHFAC effect is localized to the segment of application. As observed in the motor system, lower KHFAC frequency and higher amplitudes results in higher onset activity (Bhadra & Kilgore, 2005; Bhadra et al., 2007; Green et al., 2022).

To mitigate the effect of onset activity, different types of waveform strategies for electrical nerve block have been developed. It is possible to provide electrical nerve block using direct current by either creating a depolarization or hyperpolarization to the nerve (Bhadra & Kilgore, 2004a). However, this approach has the potential to damage the nerve if the current is applied for a prolonged period (Ackermann et al., 2011; Cogan, 2008). This damage is a result of reactive species produced at the electrode interface due to Faradaic reactions. There are several strategies that have been deployed to prevent reactive species from damaging the nerve. It is possible to coat electrodes with materials which increase the capacitive properties of the electrode. These include platinum black (Vrabec et al., 2016) and carbon ink (Goh et al., 2022). In addition to high capacitance materials, the DC waveform needs to include a recharge phase which restores the potential back to a net zero value. The recharge phase needs to be opposite and equal in charge to the block phase of the waveform (Vrabec et al., 2016). Unfortunately, during the recharge phase, the nerve is no longer blocked, which limits the total amount of time that the block can be applied. A new design using a multi-contact electrode which applies block on each contact in turn (the 'carousel' electrode) was developed to provide the ability to block the nerve continuously (Vrabec et al., 2017). The carousel electrode was used to implement paravertebral sympathetic nerve block to reduce VT inductility post-MI (Chui et al., 2017). In a porcine MI model, the carousel electrode was placed between the T1-T2 ganglia. DC nerve block was applied for up to 5 min at a time without any onset activity. A stimulation electrode at T2 was used as in previous experiments to elicit increases in heart rate and LV d$p$/d$t$ and a decrease in ARI. These perturbations were reversed with the application of DC block, with lower levels of block producing a graded effect. VT inducibility in the MI animals was tested both with and without DC block. For animals that survived the initial inducibility tests, the application of DC block prevented the reinduction of VT in all but one animal, demonstrating the cardioprotective ability of DC block of the sympathetic input.

Novel neuromodulation techniques can fine tune the nervous system for individual needs as well as provide an adaptive therapy that responds to the evolving disease state of the patient. To provide this advanced therapy, the neuromodulation device needs to have the ability to evaluate the current state of the patient and respond to perturbations. Existing and novel sensors including chemical sensing probes can be incorporated into the algorithm to monitor and respond to the patient's needs. Increasing levels of catecholamines, can be used to identify individuals at high risk for sudden cardiac death. Sympathetic block can be applied to clamp the levels of catecholamines in response to sudden increases. The use of fast scanning cyclic voltammetry (FSCV) has been demonstrated as an effective technique for measuring catecholamine levels *in vivo* and has been used to evaluate the efficacy of KHFAC on catecholamine release (Hadaya et al., 2022). Similar to previous experiments, the block electrode was placed at T1-T2 and stimulation was applied to generate a haemodynamic response. Myocardial interstitial catecholamine levels were measured using 30 cm, flexible platinum electrodes to accommodate

movement of the heart due to beating and breathing activity. FSCV was employed to measure noradrenaline. FSCV is performed by placing the electrode near the source of the transmitter and driving its potential though the oxidation/reduction potentials using a voltage-clamp circuit. The circuit working electrode potential is clamped relative to a local reference electrode. Thus, as the electrode potential is driven positive to the oxidation potential, noradrenaline is oxidized to a quinone product. The oxidation reaction generates electrons that are then measured as a compensating current in the voltage clamp and report the detection of molecules of noradrenaline. Driving the electrode potential back to a negative polarization reduces the quinone product to regenerate the catecholamine. In these experiments, FSCVs were performed both during right sympathetic chain stimulation (RSS) alone

and then with RSS and KHFAC block combined. Substantial increases in catecholamine release were measured during RSS, with variability in levels depending on the location of the probes on the surface of the heart. When KHFAC was applied preemptively to RSS, the increase in catecholamine levels was prevented.

The results of these studies provide a versatile toolbox for the treatment of cardiac dysfunction. The combination of these techniques offers the promise of an adaptable system that can address patient needs in a focused manner. There is great potential for synergy with cross-disciplinary advanced controller concepts such as fuzzy logic controllers to integrate multiple sensor inputs and neuromodulation outputs into a multi-input multi-output device design. These custom approaches can help create sustainable clinical outcomes for the most complex disease modalities.

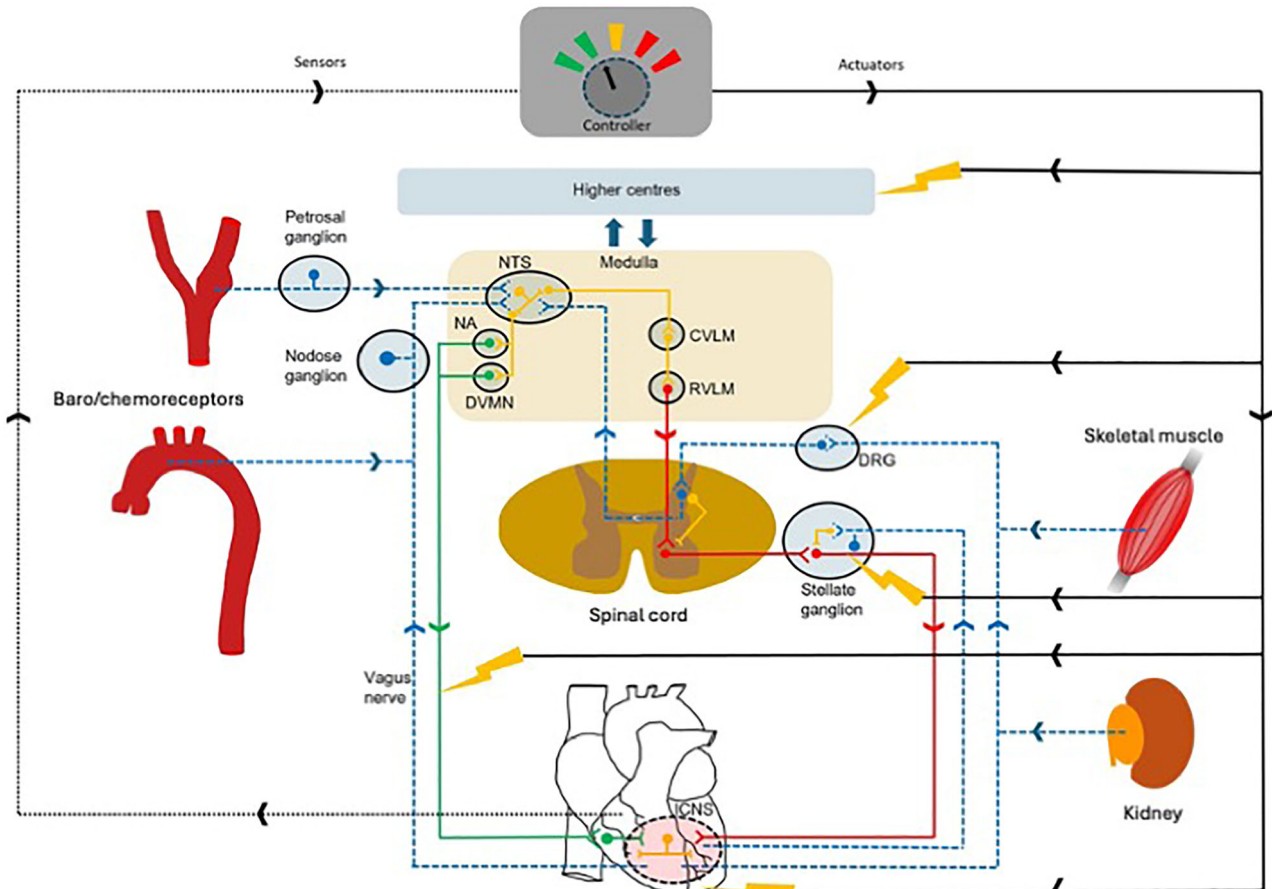

**Figure 3. Neurocardiology is based on the premise that cardiac control must be evaluated in the context of the end-organ substrate and interdependent interactions within multiple levels of the cardiac nervous system**

Imbalances in autonomic responses, while beneficial in the short term, ultimately contribute to the evolution of cardiac pathology. As our understanding of where to target emerges in terms of actuators (including the heart and intracardiac nervous system (ICNS), stellate ganglia, dorsal root ganglia (DRG), brainstem and even higher centres), there is also a need to develop sensor technology to respond to appropriate biomarkers (electrophysiological, mechanical and molecular) such that closed-loop autonomic regulation therapies can evolve. The goal is to work with endogenous control systems, rather than in opposition to them, to improve outcomes.

## Questions

**Autonomic regulation therapy for cardiovascular disease.**

- What are the preferential sites for ART in the cardiac neuraxis? Do they differ depending on the aetiology of the cardiovascular disease? How does it relate to patient selection?
- What is/are the critical neural targets for ART – afferent, efferent or local circuit neurons – intrathoracic *vs.* central?
- What are the neurotransmitter release profiles in relation to different stimulation frequencies and patterns and does this influence efficacy?
- Can closed loop systems be developed such that stimulation is adjusted to biological responses?
- How can we identify patients likely to benefit from different ARTs as they apply to different pathologies?

**Concluding perspectives.** Significant progress is being made in understanding the enormous complexity within the multiple levels of the cardiac nervous system both peripherally and centrally. An overarching framework has emerged where imbalances in autonomic evoked responses, while beneficial in the short term, are often hyper-dynamic and ultimately contribute to the progression of cardiac pathophysiology. With a deeper mechanistic understanding of these interactions, novel neural based therapeutics have emerged with the potential to restore more balanced autonomic response. However, many challenges remain particularly in identifying the best sites to target within the cardiac nervous system, at the most appropriate time in the specific disease progression and in a way that works most synergistically with current standard of care therapies. Interventions need to assessed in the context of the end-organ substrate, as well as their interactions within multiple levels of the cardiac nervous system and other neuro-humoral responses to fully understand their potential efficacy and side effects. As our understanding of where to target emerges, there is also a need to develop and refine technology to sense and respond to appropriate biomarkers such that closed-loop autonomic regulation therapies can evolve as is being pioneered by *The Leducq International Network on Bioelectronics for Neurocardiology* (Paterson et al., 2023) and to identify patients likely to benefit from the appropriate intervention. The goal is to work with endogenous control systems, rather than in opposition to them, to improve outcomes (Fig. 3).

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

## Additional information

### Competing interests

None declared.

### Author contributions

All authors contributed to the writing of the paper and approved the final version.

### Funding

NH is supported by a British Heart Foundation Senior Clinical Research Fellowship (FS/SCRF/20/32 005). AVG is supported by the BHF (RG/19/5/34 463). DJP is supported by a BHF Special Project Grant and a Leducq International Network of Excellence Award on Bioelectronics for Neurocardiology (with NH, OA, JLA and KS). JFRP is funded by the Health Research Council, Marsden Fund, the Royal Society of New Zealand and the Sidney Taylor Trust. The following NIH grant support is acknowledged: P01HL164311 (OA, JLA), R01HL162921 (JLA, CS, TV), R01HL150136 (TV, JLA, CS), NIH SPARC 1OT2OD023848 (OA, JLA), U54AT012307 (JO), R21 DK128663 (JO), R01 HL116476 (JO).

## Keywords

autonomic nervous system, cardiac function, parasympathetic, sympathetic nervous system

## Supporting information

Additional supporting information can be found online in the Supporting Information section at the end of the HTML view of the article. Supporting information files available:

**Peer Review History**

