## [Peer Review History · The Journal of Physiology]

White Paper: Neurocardiology - translational advancements and potential

Neil Herring, Olujimi A Ajijola, Robert D Foreman, Alexander V Gourine, Alexander Laurence Green, John W Osborn, David J. Paterson, Julian F. R. Paton, Crystal M Ripplinger, Corey Smith, Tina Vrabcac, Han-Jun Wang, Irving H. Zucker, and Jeffrey Laurence Ardell

DOI: 10.1113/JP284740

Corresponding author(s): Neil Herring (neil.herring@dpag.ox.ac.uk)

The following individual(s) involved in review of this submission have agreed to reveal their identity: Stavros Stavrakis (Referee #2)

Review Timeline:

Submission Date:	06-Mar-2024
Editorial Decision:	21-Mar-2024
Revision Received:	02-May-2024
Editorial Decision:	13-May-2024
Revision Received:	02-Sep-2024
Accepted:	03-Sep-2024

Senior Editor: Harold Schultz

Reviewing Editor: Harold Schultz

Transaction Report:

Dear Professor Herring,

Re: JP-WP-2024-284740 "White Paper: Translational neurocardiology: preclinical models and cardioneural integrative aspects" by Neil Herring, Olujimi A Ajjola, Robert D Foreman, Alexander V Gourine, Alexander Laurence Green, John W Osborn, David J. Paterson, Julian F. R. Paton, Crystal M Ripplinger, Corey Smith, Tina Vrabec, Han-Jun Wang, Irving H. Zucker, and Jeffrey Laurence Ardell

Thank you for submitting your manuscript to The Journal of Physiology. It has been assessed by a Reviewing Editor and by 2 expert referees and I am pleased to tell you that it is considered to be acceptable for publication following satisfactory revision.

The reports are copied at the end of this email. Please address all of the points and incorporate all requested revisions, or explain in your Response to Referees why a change has not been made.

NEW POLICY: In order to improve the transparency of its peer review process The Journal of Physiology publishes online as supporting information the peer review history of all articles accepted for publication. Readers will have access to decision letters, including all Editors' comments and referee reports, for each version of the manuscript and any author responses to peer review comments. Referees can decide whether or not they wish to be named on the peer review history document.

I hope you will find the comments helpful and have no difficulty returning revisions within 4 weeks.

Your revised manuscript should be submitted online using the links in Author Tasks Link Not Available.

If you have any queries please reply to this email and staff will be happy to assist.

Yours sincerely,

Harold Schultz
Senior Editor
The Journal of Physiology

REQUIRED ITEMS:

- Please include an Abstract Figure file, as well as the Figure Legend text within the main article file. The Abstract Figure is a piece of artwork designed to give readers an immediate understanding of the Review Article and should summarise the main conclusions. If possible, the image should be easily 'readable' from left to right or top to bottom. It should show the physiological relevance of the Review so readers can assess the importance and content of the article. Abstract Figures should not merely recapitulate other figures in the Review. Please try to keep the diagram as simple as possible and without superfluous information that may distract from the main conclusion of the Review. Abstract Figures must be provided by authors no later than the revised manuscript stage and should be uploaded as a separate file during online submission labelled as File Type 'Abstract Figure'. Please ensure that you include the figure legend in the main article file. All Abstract Figures will be sent to a professional illustrator for redrawing and you may be asked to approve the redrawn figure before your paper is accepted.

- Your MS must include a complete "Additional information section" with the following 4 headings and content:

Competing Interests: A statement regarding competing interests. If there are no competing interests, a statement to this effect must be included. All authors should disclose any conflict of interest in accordance with journal policy.

Author contributions: Each author should take responsibility for a particular section of the study and have contributed to writing the paper. Acquisition of funding, administrative support or the collection of data alone does not justify authorship; these contributions to the study should be listed in the Acknowledgements. Additional information such as 'X and Y have contributed equally to this work' may be added as a footnote on the title page.

It must be stated that all authors approved the final version of the manuscript and that all persons designated as authors qualify for authorship, and all those who qualify for authorship are listed.

Funding: Authors must indicate all sources of funding, including grant numbers. If authors have not received funding, this

must be stated.

It is the responsibility of authors funded by RCUK to adhere to their policy regarding funding sources and underlying research material. The policy requires funding information to be included within the acknowledgement section of a paper. Guidance on how to acknowledge funding information is provided by the Research Information Network. The policy also requires all research papers, if applicable, to include a statement on how any underlying research materials, such as data, samples or models, can be accessed. However, the policy does not require that the data must be made open. If there are considered to be good or compelling reasons to protect access to the data, for example commercial confidentiality or legitimate sensitivities around data derived from potentially identifiable human participants, these should be included in the statement.

Acknowledgements: Acknowledgements should be the minimum consistent with courtesy. The wording of acknowledgements of scientific assistance or advice must have been seen and approved by the persons concerned. This section should not include details of funding.

- Please upload separate high quality figure files via the submission form.

- Author profile(s) must be uploaded via the submission form. Authors should submit a short biography (no more than 100 words for one author or 150 words in total for two authors) and a portrait photograph of the two leading authors on the paper. These should be uploaded and clearly labelled together in a Word document with the revised version of the manuscript. Any standard image format for the photograph is acceptable, but the resolution should be at least 300 DPI and preferably more. A group photograph of all authors is also acceptable, providing the biography for the whole group does not exceed 150 words.

EDITOR COMMENTS

Reviewing Editor:

Please respond to reviewer comments. Please also see 'Required Items' above.

Senior Editor:

Thank you for submission of your white paper article to the Journal of Physiology for consideration. The article has been reviewed by experts in the field and found to be potentially acceptable for publication pending adequate revision to address all of the concerns raised. Referee 1 made several points that are viewed to be important to address. Specific comments suggest that in some cases, new additions to the literature that have evolved in the field since 2016 were omitted or not sufficiently discussed, particularly neuro-modulation and measurement strategies in humans. Whereas, it was thought that as an update to a white paper, some older data that have not advanced or proved promising since 2016 may be over emphasized. Additional figures are welcome to assist with emphasizing or clarifying important points. Please address all comments from the external referees as well as addressing the list of requirements or publication of a review in the journal.

REFEREE COMMENTS

Referee #1:

This is a white paper review by Herring et al. on translational aspects of neurocardiology. It is an update from 2016 and as such it is an important point of reference for all those interested in this ever-expanding discipline.

Comments:

This area is of immense importance and this update will provide the readership an authoritative review.

Major comments:

1. Abstract - please add some more phrases in terms of the key updates in this field since 2016.
2. In general - in the introduction it is important to define the recent advances in neurocardiology since the last white paper in 2016. Perhaps the text can also focus on this as an update rather than a recapitulation of what was previously in the white document. The reader can simply be referred to the past document for details.
 - a. The text should end with imminent areas where the field can translate to the bedside and also future avenues that the authors (pre-eminent in their field) envisage the highest yields in terms of research questions.
 - b. What was envisaged to have been of immense clinical value has not, since 2016, shown as much promise and it is important for the veritable experts (authors listed here) to emphasise these changes within this document.
3. It is important that this review has begun with the role of the afferents in cardiac health and disease.
 - a. Page 4 line 107: there is central remodelling, but work from Zucker, an esteemed author on this review, has also shown loss of arborisation of atrial afferent (stretch) receptors at the level of the heart and this correlates with alterations in atrial compliance. Though this is presented later, this may be worthwhile introducing in this paragraph.
 - b. Perhaps this section could be broken into atrial components and ventricular components. A discussion of autonomic remodelling in the atria and then autonomic remodelling in the ventricle - with the implications of each work then presented.
 - c. Also - the physiology is presented in section 1 and then perhaps the remodelling work could be presented later in a section on autonomic dysfunction related to cardiovascular disease, followed by the section on neuromodulation.
4. There are only 2 figures - is this deliberate? This seems to be a major drawback to this otherwise authoritative review.
5. Line 64: are the authors referring to predominantly ventricular receptors? Please clarify
6. The section on afferents will immensely benefit from a discussion of atrial receptors that respond to atrial stretch (see the reviews by Thoren (<https://www.ncbi.nlm.nih.gov/pubmed/386467>), Hainsworth- already referenced in this paper) as well as historic work from Nonidez, Paintal, Ledsome and Linden, Kapagoda and many others.
7. There has already been important human data showing that there are deficits in the reflex responses of decreased blood volume to the heart (without changes in arterial pressure) that are presumed to be the result of deficiencies in the function of cardiopulmonary receptors in several disease states.
 - a. Alcohol (Narkiewicz K, Cooley RL, Somers VK. Alcohol potentiates orthostatic hypotension: implications for alcohol-related syncope. *Circulation*. 2000;101:398-402)
 - b. Heart failure (Mohanty PK, Arrowood JA, Ellenbogen KA, Thames MD. Neurohumoral and hemodynamic effects of lower body negative pressure in patients with congestive heart failure. *Am Heart J*. 1989;118:78-85.
 - c. Most recently significant deficiencies have been seen in patients with Atrial fibrillation (a heart rhythm disorder characterised by irregular heartbeats with implications for baroreflex function) both in normal and abnormal rhythm: Malik V, McKittrick DJ, Lau DH, Sanders P, Arnold LF. Clinical evidence of autonomic dysfunction due to atrial fibrillation: implications for rhythm control strategy. *J Interv Card Electrophysiol*. 2019;54:299-307.
 - d. Malik V, Elliott AE, Thomas G, et al. Autonomic afferent dysregulation in atrial fibrillation. *J Am Coll Cardiol EP*. 2022;8(2), 152-164
 - e. (Animal studies which have shown receptor abnormalities in a canine model of heart failure with histologic characteristics similar to those described by Nonidez, Paintal and Coleridge (from atrial tissue -particularly pulmonary-vein atrial junctions) - see Zucker et al. *J Clin Invest* 1977 have already been discussed). Though the study described had also assessed these neural changes in relation to atrial compliance - a critical point, which speaks to the role of atrial remodeling and atrial stretch - which has not been considered here.
8. There is a nice introduction to the concept of interoception and some top-down processing in autonomic disorders that produce alterations in the heart (arrhythmia etc).
9. Page 28 and also page 32- a diagram here outlining the pathophysiology of CSAR (promoting VT) would be helpful. See

<https://www.ncbi.nlm.nih.gov/pubmed/37855773> for an example of this. It would also nicely preface the discussion on TRPV1 and RTX which has been emphasised here.

10. Cardiac arrhythmia (page 28) should be broken down to atrial arrhythmia (AF especially has had several important updates - Jayachandran et al. Circ 2000; Chang et al. Circ 2001; Gussak et al. JCI Insight 2019; Yu et al. PACE 2014; Chen et al. Circ Res 2014; Wasmund et al. Circ 2003 in addition to the work on volume baroreflexes and AF described above) and VT/VF. At least in AF- the role of the efferents has been the focus of prior work -with little attention given to afferent nerves.

11. There are several areas in the text that could be simplified as they are too long- some of the questions in particular, could be refined to make short sentences with no more than 5 questions per box. Another area is from page 35- a brief introduction in other disorders is acceptable -but this section is tangential and needs to focus on neurocardiology.

a. The blood pressure treatments described from 1998 on the RVLM have not made it into the clinical arena. Are these still relevant to mention? In general, this section is very much a missed opportunity in terms of what could be discussed and explored in regard to the current, exciting literature. The first page and a half of this section contains no information regarding current therapeutics in cardiac neuromodulation.

b. There is an over-emphasis of the VNS and heart failure studies. Though no formal publication is available - the results were presented in 2023 and negative. Nectar-HF was also negative for the primary (and secondary) endpoints though there may have been some improvements in quality of life.

c. Clearer demarcation of clinical work from pre-clinical (and hypothesis generating work) is needed in this section. It is creating confusion as to what is thought to be "standard" and well- accepted vs postulating arguments in favour of something.

d. Page 40 line 1120- I think this section is a good opportunity to highlight some important clinical studies on neuromodulation and POTS as well as AF by Stavrakis and colleagues. I do not agree with the statement that this is a bridge to a more "definitive" intervention in invasive vagal nerve stimulation. There has not been any significant work on invasive VNS and AF reduction in these patients. Further, the mechanism of action of low-level VNS or LLTS- (though less clearly defined) are likely anti-adrenergic and anti-inflammatory and thus quite different to the effect of invasive vagal nerve stimulation. There is clear chronicity in the effects of LLTS- in TREAT AF, in some individuals with AF, there was a burden reduction in AF at 6 months with just 1 hour per day of LLTS. Some anti-adrenergic data (on the stellate) are later presented - but incoherent.

e. Several parameters for VNS are yet to be studied -but the data thus far are disappointing. It is no longer as relevant to emphasise these in emerging treatments and the recent negative data needs to be made clear.

f. Page 1165 -to state that SCS has a 20-year history in refractory angina implies that it has been standard of care for 20 years and this is not the case. It is better to state that SCS was considered 20 years ago for angina. However - more importantly for angina (and also highly relevant for VT/VF) is stellectomy/sympathectomy for angina (see <https://www.ncbi.nlm.nih.gov/pubmed/37855773>) was considered 100 years ago...

g. Stellectomy is established in the guidelines for arrhythmias such as CPVT and LQTS and this is later discussed in the manuscript (with work appearing in another related white paper).

h. The role of stellate ganglion blockade, however, has not been mentioned in this section on neurocardiology. This is an omission given the increased clinical traction world-wide in the temporising of patients with refractory ventricular arrhythmia. See <https://doi.org/10.1093/eurheartj/ehae083> for a recent overview.

i. Also, procedures such as ganglionated plexus ablation (with several preclinical and clinical studies) as well as cardioneuroablation for vasovagal syncope could be discussed as this section (which is too long) is refined.

12. It would be well worth discussing the work done by the Macefield group on vagal nerve measurements in humans using micro-electrodes- this is likely to provide insight in humans, on several links between the ANS and the heart- hitherto uncovered.

Minor comments:

1. Page 28 line 774 - other major recent reviews regarding this should be referenced (Chen et al Circ Res 2014 especially) and Linz et al IJC 2019.

a. Chen et al. has a seminal figure with the electrophysiologic role of the efferent ANS detailed and its proarrhythmic proclivity.

Referee #2:

This is an update of the original white paper on translational neurocardiology, published in the Journal of Physiology in 2016. This is well written and comprehensive review.

Comments:

1. The anti-inflammatory effect of VNS and tragus stimulation in HFpEF (both clinical and experimental) would be an important addition to the manuscript.
2. The importance of patient selection for ART should be mentioned, as it applies to both atrial and ventricular arrhythmias and HF.
3. Non-invasive spinal cord stimulation should also be mentioned as a more clinically applicable modality for ART
4. Page 35, line 973-4: "What is the best approach to control clinical sympathetic storms to reduce blood pressure 974 spikes and prevent serious adverse events?". This question as it is written implies that controlling sympathetic storms can only be accomplished by reducing blood pressure. It can be broken into 2 bullet points, to separate the control of sympathetic storms from the control of blood pressure.

END OF COMMENTS

Confidential Review

06-Mar-2024

REQUIRED ITEMS:

- Please include an Abstract Figure file, as well as the Figure Legend text within the main article file. The Abstract Figure is a piece of artwork designed to give readers an immediate understanding of the Review Article and should summarise the main conclusions. If possible, the image should be easily 'readable' from left to right or top to bottom. It should show the physiological relevance of the Review so readers can assess the importance and content of the article. Abstract Figures should not merely recapitulate other figures in the Review. Please try to keep the diagram as simple as possible and without superfluous information that may distract from the main conclusion of the Review. Abstract Figures must be provided by authors no later than the revised manuscript stage and should be uploaded as a separate file during online submission labelled as File Type 'Abstract Figure'. Please ensure that you include the figure legend in the main article file. All Abstract Figures will be sent to a professional illustrator for redrawing and you may be asked to approve the redrawn figure before your paper is accepted.

Response: Now included.

- Your MS must include a complete "Additional information section" with the following 4 headings and content:

Competing Interests: A statement regarding competing interests. If there are no competing interests, a statement to this effect must be included. All authors should disclose any conflict of interest in accordance with journal policy.

Author contributions: Each author should take responsibility for a particular section of the study and have contributed to writing the paper. Acquisition of funding, administrative support or the collection of data alone does not justify authorship; these contributions to the study should be listed in the Acknowledgements. Additional information such as 'X and Y have contributed equally to this work' may be added as a footnote on the title page.

It must be stated that all authors approved the final version of the manuscript and that all persons designated as authors qualify for authorship, and all those who qualify for authorship are listed.

Funding: Authors must indicate all sources of funding, including grant numbers. If authors have not received funding, this must be stated.

It is the responsibility of authors funded by RCUK to adhere to their policy regarding funding sources and underlying research material. The policy requires funding information to be included within the acknowledgement section of a paper. Guidance on how to acknowledge funding information is provided by the Research Information Network. The policy also requires all research papers, if applicable, to include a statement on how any underlying research materials, such as data, samples or models,

can be accessed. However, the policy does not require that the data must be made open. If there are considered to be good or compelling reasons to protect access to the data, for example commercial confidentiality or legitimate sensitivities around data derived from potentially identifiable human participants, these should be included in the statement.

Acknowledgements: Acknowledgements should be the minimum consistent with courtesy. The wording of acknowledgements of scientific assistance or advice must have been seen and approved by the persons concerned. This section should not include details of funding.

Response: Additional information sections now included.

- Please upload separate high quality figure files via the submission form.

Response: now uploaded.

- Author profile(s) must be uploaded via the submission form. Authors should submit a short biography (no more than 100 words for one author or 150 words in total for two authors) and a portrait photograph of the two leading authors on the paper. These should be uploaded and clearly labelled together in a Word document with the revised version of the manuscript. Any standard image format for the photograph is acceptable, but the resolution should be at least 300 DPI and preferably more. A group photograph of all authors is also acceptable, providing the biography for the whole group does not exceed 150 words.

Response: This has now been included for the group.

Referee #1

This area is of immense importance and this update will provide the readership an authoritative review.

Major comments:

1. Abstract - please add some more phrases in terms of the key updates in this field since 2016.

Response: We have expanded the abstract and also added a graphical abstract figure and summary Figure 3 to highlight where we feel the field has advanced and is going in the future.

2. In general - in the introduction it is important to define the recent advances in neurocardiology since the last white paper in 2016. Perhaps the text can also focus on this as an update rather than a recapitulation of what was previously in the white document. The reader can simply be referred to the past document for details.-

Response: This is a good point and we have now made this clear in our introduction.

a. The text should end with imminent areas where the field can translate to the bedside and also future avenues that the authors (pre-eminent in their field) envisage the highest yields in terms of research questions.

Response: There are many areas that need addressing in the field to help with understanding and translation and we have tried to highlight these in the grey boxes throughout the text, as we did in the white paper in 2016. We have revised these as suggested. We have tried to take a broader perspective in the conclusion highlighting how future approaches to interacting with the ANS should be implemented (eg. reversibility, sensor driven closed loop systems, working with reflex responses rather than against them). We have now included figure 3 emphasizing this overall approach as well.

b. What was envisaged to have been of immense clinical value has not, since 2016, shown as much promise and it is important for the veritable experts (authors listed here) to emphasise these changes within this document.

Response: We agree that autonomic regulation therapy has not become part of mainstream practice and have now acknowledged this up front in the introduction. Showing “promise” is subjective opinion, and we feel that there remains significant translational potential highlighted here and in the accompanying other two white papers.

3. It is important that this review has begun with the role of the afferents in cardiac health and disease.

a. Page 4 line 107: there is central remodelling, but work from Zucker, an esteemed author on this review, has also shown loss of arborisation of atrial afferent (stretch) receptors at the level of the heart and this correlates with alterations in atrial compliance. Though this is presented later, this may be worthwhile introducing in this paragraph. –

Response: Thank you. This is mentioned later in the section on “*The relevance of the cardiac neuronal hierarchy to heart failure*”, however we have now referenced this earlier work as suggested at this earlier stage.

b. Perhaps this section could be broken into atrial components and ventricular components. A discussion of autonomic remodelling in the atria and then autonomic remodelling in the ventricle - with the implications of each work then presented.

Response: We have added a paragraph concerning differences in atrial vs ventricular reflexes to the end of this section as advised.

c. Also - the physiology is presented in section 1 and then perhaps the remodelling work

could be presented later in a section on autonomic dysfunction related to cardiovascular disease, followed by the section on neuromodulation. –

Response: As requested by the reviewer (point 3a), we have added pertinent points regarding specific remodelling in this physiology section and feel that this is the best way of presenting.

4. There are only 2 figures - is this deliberate? This seems to be a major drawback to this otherwise authoritative review.

Response: We have now added a third main figure and an abstract figure.

5. Line 64: are the authors referring to predominantly ventricular receptors? Please clarify

Response: This refers to both atrial and ventricular innervation so we have reworded to clarify here.

6. The section on afferents will immensely benefit from a discussion of atrial receptors that respond to atrial stretch (see the reviews by Thoren (<https://www.ncbi.nlm.nih.gov/pubmed/386467> [ncbi.nlm.nih.gov]), Hainsworth- already referenced in this paper) as well as historic work from Nonidez, Paintal, Ledson and Linden, Kapagoda and many others.

Response: We have now referenced the Thoren review from 1979 regarding atrial stretch receptors which highlights the classical work in the area. As the white paper tries to focus on advances since 2016, we have not referenced all of the primary historical work given that this is beyond the remit of the review.

7. There has already been important human data showing that there are deficits in the reflex responses of decreased blood volume to the heart (without changes in arterial pressure) that are presumed to be the result of deficiencies in the function of cardiopulmonary receptors in several disease states.

a. Alcohol (Narkiewicz K, Cooley RL, Somers VK. Alcohol potentiates orthostatic hypotension: implications for alcohol-related syncope. *Circulation*. 2000;101:398-402)

b. Heart failure (Mohanty PK, Arrowood JA, Ellenbogen KA, Thames MD. Neurohumoral and hemodynamic effects of lower body negative pressure in patients with congestive heart failure. *Am Heart J*. 1989;118:78-85.

c. Most recently significant deficiencies have been seen in patients with Atrial fibrillation (a heart rhythm disorder characterised by irregular heartbeats with implications for baroreflex function) both in normal and abnormal rhythm: Malik V, McKittrick DJ, Lau DH, Sanders P, Arnold LF. Clinical evidence of autonomic dysfunction due to atrial fibrillation: implications for rhythm control strategy. *J Interv Card Electrophysiol*.

2019;54:299-307.

d. Malik V, Elliott AE, Thomas G, et al. Autonomic afferent dysregulation in atrial fibrillation. J Am Coll Cardiol EP. 2022;8(2), 152-164

e. (Animal studies which have shown receptor abnormalities in a canine model of heart failure with histologic characteristics similar to those described by Nonidez, Paintal and Coleridge (from atrial tissue -particularly pulmonary-vein atrial junctions) - see Zucker et al. J Clin Investig 1977 have already been discussed). Though the study described had also assessed these neural changes in relation to atrial compliance - a critical point, which speaks to the role of atrial remodeling and atrial stretch - which has not been considered here.

Response:

We agree and have added a statement citing some of this human research as suggested.

8. There is a nice introduction to the concept of interoception and some top-down processing in autonomic disorders that produce alterations in the heart (arrhythmia etc).

9. Page 28 and also page 32- a diagram here outlining the pathophysiology of CSAR (promoting VT) would be helpful. See <https://www.ncbi.nlm.nih.gov/pubmed/37855773> [[ncbi.nlm.nih.gov](https://www.ncbi.nlm.nih.gov)] for an example of this. It would also nicely preface the discussion on TRPV1 and RTX which has been emphasised here.

Response: This article is in press, with a preview of the corrected proof only available online. We would happily consider incorporating or modifying this once it is publically available.

10. Cardiac arrhythmia (page 28) should be broken down to atrial arrhythmia (AF especially has had several important updates - Jayachandran et al. Circ 2000; Chang et al. Circ 2001; Gussak et al. JCI Insight 2019; Yu et al. PACE 2014; Chen et al. Circ Res 2014; Wasmund et al. Circ 2003 in addition to the work on volume baroreflexes and AF described above) and VT/VF. At least in AF- the role of the efferents has been the focus of prior work -with little attention given to afferent nerves.

Response: We have added the suggested references with regards to AF as they form an interesting parallel between autonomic remodelling seen in the ventricle following MI and heart failure. The overall aim though is for an overview of general arrhythmogenic principles, with the accompanying clinical white paper focusing more specifically on types of arrhythmia to avoid excessive duplication.

11. There are several areas in the text that could be simplified as they are too long- some of the questions in particular, could be refined to make short sentences with no more than 5 questions per box. Another area is from page 35- a brief introduction in

other disorders is acceptable -but this section is tangential and needs to focus on neurocardiology.

Response: We have reduced the number of questions at the end of each section. We have also cut down the section on deep brain stimulation to keep it focused on the autonomic effects.

a. The blood pressure treatments described from 1998 on the RVLM have not made it into the clinical arena. Are these still relevant to mention? In general, this section is very much a missed opportunity in terms of what could be discussed and explored in regard to the current, exciting literature. The first page and a half of this section contains no information regarding current therapeutics in cardiac neuromodulation.

Response: We have now removed the section on the RVLM. In this section we try to focus on where translation has failed and how approaches could be changed in the future, but please be aware that an entirely separate clinical white paper will accompany this one where more emphasis is placed on clinical studies and current therapeutics.

b. There is an over-emphasis of the VNS and heart failure studies. Though no formal publication is available - the results were presented in 2023 and negative. Nectar-HF was also negative for the primary (and secondary) endpoints though there may have been some improvements in quality of life.

Response: Several authors on this white paper are chief investigators for for ANTHEM HFrEF Pivotal. Data has been presented at Heart Rhythm (New Orleans) in 2023 (NH) and further data will be presented at INS (Vancouver) 2024 (JA). Despite the study finishing early, this was not on the grounds of efficacy or safety. Whilst the results are not yet published, we can reassure the reviewer that it is not an entirely negative study as suggested. We also feel that VNS continues to hold promise in many different forms and the failure of previous studies are important learning point with regards to translation.

c. Clearer demarcation of clinical work from pre-clinical (and hypothesis generating work) is needed in this section. It is creating confusion as to what is thought to be "standard" and well- accepted vs postulating arguments in favour of something.

Response: This is a good point and we have reworded this section to make this clear throughout.

d. Page 40 line 1120- I think this section is a good opportunity to highlight some important clinical studies on neuromodulation and POTS as well as AF by Stavarakis and colleagues. I do not agree with the statement that this is a bridge to a more "definitive" intervention in invasive vagal nerve stimulation. There has not been any significant work on invasive VNS and AF reduction in these patients. Further, the mechanism of action of low-level VNS or LLTS- (though less clearly defined) are likely anti-adrenergic and anti-inflammatory and thus quite different to the effect of invasive vagal nerve

stimulation. There is clear chronicity in the effects of LLTS- in TREAT AF, in some individuals with AF, there was a burden reduction in AF at 6 months with just 1 hour per day of LLTS. Some anti-adrenergic data (on the stellate) are later presented - but incoherent.

Response: This is a good point and we have removed the comment to tragus stimulation being a “bridge” and included it as a potential treatment for heart failure in this section as well as discussing it in the context of AF. This is also covered in more detail in the clinical white paper where Dr Stavrakis is a co-author and we have also cross referenced this.

e. Several parameters for VNS are yet to be studied -but the data thus far are disappointing. It is no longer as relevant to emphasise these in emerging treatments and the recent negative data needs to be made clear.

Response: We agree that (published) clinical data to date have been disappointing, and we have now made this clear. In our opinion though, interacting with the vagus nerve is still an important translational avenue and remains relevant.

f. Page 1165 -to state that SCS has a 20-year history in refractory angina implies that it has been standard of care for 20 years and this is not the case. It is better to state that SCS was considered 20 years ago for angina. However - more importantly for angina (and also highly relevant for VT/VF) is stellectomy/sympathectomy for angina (see <https://www.ncbi.nlm.nih.gov/pubmed/37855773> [[ncbi.nlm.nih.gov](https://www.ncbi.nlm.nih.gov/pubmed/37855773)]) was considered 100 years ago...

Response: We agree that this could be misleading and have reworded to remove any ambiguity.

g. Stellectomy is established in the guidelines for arrhythmias such as CPVT and LQTS and this is later discussed in the manuscript (with work appearing in another related white paper).

Response: We agree and we have now highlighted this and flagged that it discussed in more detail in the third clinical white paper accompanying this one.

h. The role of stellate ganglion blockade, however, has not been mentioned in this section on neurocardiology. This is an omission given the increased clinical traction world-wide in the temporising of patients with refractory ventricular arrhythmia. See <https://doi.org/10.1093/eurheartj/ehae083> [[doi.org](https://doi.org/10.1093/eurheartj/ehae083)] for a recent overview.

Response: This is a good point and we have added the suggested reference.

i. Also, procedures such as ganglionated plexus ablation (with several preclinical and clinical studies) as well as cardioneuroablation for vasovagal syncope could be discussed as this section (which is too long) is refined.

Response: We have avoided discussing this to prevent duplication given it is included in the Clinical White paper and it to prevent this section becoming even longer.

12. It would be well worth discussing the work done by the Macefield group on vagal nerve measurements in humans using micro-electrodes- this is likely to provide insight in humans, on several links between the ANS and the heart- hitherto uncovered.

Response: We agree and have now referenced this work.

Minor comments:

1. Page 28 line 774 - other major recent reviews regarding this should be referenced (Chen et al Circ Res 2014 especially) and Linz et al IJC 2019.

a. Chen et al. has a seminal figure with the electrophysiologic role of the efferent ANS detailed and its proarrhythmic proclivity.

Response: The Chen review pre-dates even the previous white paper and so we have not included this as a “recent” update, which (like the figure) relates specifically to AF. We have added the more up to date Linz reference as suggested, although there are many more examples that could be included.

Referee #2:

This is an update of the original white paper on translational neurocardiology, published in the Journal of Physiology in 2016. This is well written and comprehensive review.

1. The anti-inflammatory effect of VNS and tragus stimulation in HFpEF (both clinical and experimental) would be an important addition to the manuscript.

Response: We agree and we have highlighted and referenced both an extensive recent review (Bazoukis et al 2023) and the recently reported ANTHEM-HFpEF study.

2. The importance of patient selection for ART should be mentioned, as it applies to both atrial and ventricular arrhythmias and HF.

Response: We agree and have made this a key question at the end of the ART section and highlighted this in our conclusion.

3. Non-invasive spinal cord stimulation should also be mentioned as a more clinically applicable modality for ART

Response: We agree and have now included this in the appropriate section.

4. Page 35, line 973-4: "What is the best approach to control clinical sympathetic storms

to reduce blood pressure 974 spikes and prevent serious adverse events?". This question as it is written implies that controlling sympathetic storms can only be accomplished by reducing blood pressure. It can be broken into 2 bullet points, to separate the control of sympathetic storms from the control of blood pressure.

Response: This is good point and we have reworded this accordingly.

Dear Dr Herring,

Re: JP-WP-2024-284740R1 "White Paper: Neurocardiology - translational advancements and potential" by Neil Herring, Olujimi A Ajijola, Robert D Foreman, Alexander V Gourine, Alexander Laurence Green, John W Osborn, David J. Paterson, Julian F. R. Paton, Crystal M Ripplinger, Corey Smith, Tina Vrabec, Han-Jun Wang, Irving H. Zucker, and Jeffrey Laurence Ardell

Thank you for submitting your manuscript to The Journal of Physiology. It has been assessed by a Reviewing Editor and by 2 expert referees and I am pleased to tell you that it is considered to be acceptable for publication following satisfactory revision.

The reports are copied at the end of this email. Please address all of the points and incorporate all requested revisions, or explain in your Response to Referees why a change has not been made.

NEW POLICY: In order to improve the transparency of its peer review process The Journal of Physiology publishes online as supporting information the peer review history of all articles accepted for publication. Readers will have access to decision letters, including all Editors' comments and referee reports, for each version of the manuscript and any author responses to peer review comments. Referees can decide whether or not they wish to be named on the peer review history document.

I hope you will find the comments helpful and have no difficulty returning revisions within 4 weeks.

Your revised manuscript should be submitted online using the links in Author Tasks Link Not Available.

If you have any queries please reply to this email and staff will be happy to assist.

Yours sincerely,

Harold Schultz
Senior Editor
The Journal of Physiology

EDITOR COMMENTS

Reviewing Editor:

No further comments.

Senior Editor:

Thank you for submission of your revised white paper article to the Journal of Physiology for consideration. The article has been reviewed by the external reviewers found to be acceptable for publication. However, it was brought to our attention that several areas of text represent self duplication from the prior white paper from the authors (<https://doi.org/10.1113/JP271869>). This must be addressed to prevent the appearance of self plagiarism and triggering plagiarism servers. There is no concern about the content. However, the text must be revised to prevent exact replication of sentences and in most cases complete or almost complete replication of paragraphs. Cross check the manuscript with the prior published white paper. To assist with the revision, we have listed the affected areas. Please also review the iThenticate document (**PLEASE SEE NOTE FROM EDITORIAL OFFICE BELOW**). Please submit the revised manuscript and a copy with changes highlighted in red text and so named.

Pg 13: para starting line 357, 372

Pg 17: 458

Pg 28: 775

Pg 37: 1032, 1048

Pg 38: 1081

Pg 39: 1097

Pg 40: 1134

Pg 41: 1165

Pg 42: 1179, 1189

Pg 43: 1209, 1222

pg 57: 1615

****NOTE FROM EDITORIAL OFFICE****

re: Ithenticate (CrossCheck) report:

The Ithenticate report is an online tool which shows specific areas of overlap with published content online; I'm not sure that you will be able access it in the same way that we can, at the office.

This is the link to the report: https://api.ithenticate.com/en_us/dv/20220511?lang=en_us&o=108160056

If that link doesn't work for you (and it may not; you might need an authorised Ithenticate account to see it), then please just follow the advice of the Senior Editor above as best you can, to avoid duplicating sections of text between the current submission and your earlier White Paper (<https://physoc.onlinelibrary.wiley.com/doi/10.1113/JP271869>).

REFeree COMMENTS

Referee #1:

The authors have sufficiently addressed reviewer comments- which have greatly enhanced this manuscript.

Referee #2:

I have no further comments.

END OF COMMENTS

1st Confidential Review

02-May-2024

Senior Editor:

Thank you for submission of your revised white paper article to the *Journal of Physiology* for consideration. The article has been reviewed by the external reviewers found to be acceptable for publication. However, it was brought to our attention that several areas of text represent self duplication from the prior white paper from the authors

Response: We have written all of the sections you have brought to our attention, which are highlighted in the revised manuscript with tracked changes enabled to show the rewording. We apologies for the delay in returning the manuscript.

Dear Dr Herring,

Re: JP-WP-2024-284740R2 "White Paper: Neurocardiology - translational advancements and potential" by Neil Herring, Olujimi A Ajjola, Robert D Foreman, Alexander V Gourine, Alexander Laurence Green, John W Osborn, David J. Paterson, Julian F. R. Paton, Crystal M Ripplinger, Corey Smith, Tina Vrabec, Han-Jun Wang, Irving H. Zucker, and Jeffrey Laurence Ardell

We are pleased to tell you that your paper has been accepted for publication in The Journal of Physiology.

Authors should note that it is too late at this point to offer corrections prior to proofing. Major corrections at proof stage, such as changes to figures, will be referred to the Editors for approval before they can be incorporated. Only minor changes, such as to style and consistency, should be made at proof stage. Changes that need to be made after proof stage will usually require a formal correction notice.

All queries at proof stage should be sent to: TJP@wiley.com

If you would like to receive our 'Research Roundup', a monthly newsletter highlighting the cutting-edge research published in The Physiological Society's family of journals (The Journal of Physiology, Experimental Physiology and Physiological Reports), please click this link, fill in your name and email address and select 'Research Roundup':
<https://www.physoc.org/journals-and-media/membernews/>

Yours sincerely,

Harold Schultz
Senior Editor
The Journal of Physiology

P.S. - You can help your research get the attention it deserves! Check out Wiley's free Promotion Guide for best-practice recommendations for promoting your work at www.wileyauthors.com/eoo/guide. You can learn more about Wiley Editing Services which offers professional video, design, and writing services to create shareable video abstracts, infographics, conference posters, lay summaries, and research news stories for your research at www.wileyauthors.com/eoo/promotion.

IMPORTANT NOTICE ABOUT OPEN ACCESS: To assist authors whose funding agencies mandate public access to published research findings sooner than 12 months after publication The Journal of Physiology allows authors to pay an Open Access (OA) fee to have their papers made freely available immediately on publication.

You can check if your funder or institution has a Wiley Open Access Account here: <https://authorservices.wiley.com/author-resources/Journal-Authors/licensing-and-open-access/open-access/author-compliance-tool.html>

EDITOR COMMENTS:

The editors wish to thank the authors for these final adjustments to the manuscript. The article is now accepted for

publication. Congratulations for an interesting and insightful review. Please consider the Journal of Physiology for your future studies.

2nd Confidential Review

02-Sep-2024